# Cryo-EM structure and dynamics of the green-light absorbing proteorhodopsin

Stephan Hirschi[1], David Kalbermatter [1], Zöhre Ucurum[1], Thomas Lemmin[2,3 ✉] & Dimitrios Fotiadis [1 ✉]

The green-light absorbing proteorhodopsin (GPR) is the archetype of bacterial light-driven proton pumps. Here, we present the 2.9 Å cryo-EM structure of pentameric GPR, resolving important residues of the proton translocation pathway and the oligomerization interface. Superposition with the structure of a close GPR homolog and molecular dynamics simulations reveal conformational variations, which regulate the solvent access to the intra- and extracellular half channels harbouring the primary proton donor E109 and the proposed proton release group E143. We provide a mechanism for the structural rearrangements allowing hydration of the intracellular half channel, which are triggered by changing the protonation state of E109. Functional characterization of selected mutants demonstrates the importance of the molecular organization around E109 and E143 for GPR activity. Furthermore, we present evidence that helices involved in the stabilization of the protomer interfaces serve as scaffolds for facilitating the motion of the other helices. Combined with the more constrained dynamics of the pentamer compared to the monomer, these observations illustrate the previously demonstrated functional significance of GPR oligomerization. Overall, this work provides molecular insights into the structure, dynamics and function of the proteorhodopsin family that will benefit the large scientific community employing GPR as a model protein.

[1] Institute of Biochemistry and Molecular Medicine, University of Bern, Bern, Switzerland. [2] DS3Lab, System Group, Department of Computer Sciences, ETH Zurich, Zürich, Switzerland. [3] Trkola Group, Institute of Medical Virology, University of Zurich, Zürich, Switzerland. ✉email: thomas.lemmin@inf.ethz.ch; dimitrios.fotiadis@ibmm.unibe.ch

Microbial rhodopsins are a family of highly abundant light-driven ion pumps and photoreceptors, found in a great variety of phototrophic microorganisms spanning all domains of life[1]. Retinal-based rhodopsins and chlorophyll-containing photochemical reaction centres are the two main light-harvesting mechanisms on earth, with rhodopsin genes being about three times more prevalent[2]. Among microbial rhodopsins, proteorhodopsins represent the most abundant family, contributing significantly to the global energy cycle by capturing solar energy in the ocean[3]. They are classified according to their spectral properties, which reflect the local light availability in their native environments. Green-light absorbing proteorhodopsins (GPR), with an absorption maximum at ~525 nm, and blue-light absorbing proteorhodopsins (BPR) with an absorption maximum at ~490 nm, are preferentially found near the surface of ocean waters or in subjacent layers, respectively[4]. GPR was discovered in marine γ-proteobacteria 20 years ago as the first bacterial rhodopsin and was characterized as a light-driven proton pump[5]. Since its discovery, GPR has become the archetype of bacterial proton pumps, serving as a model protein for the development of biophysical techniques, for the elucidation of protein structure-function dynamics and for applications in bionanotechnology[6–13]. Furthermore, ion-pumping microbial rhodopsins have become the basis for the engineering of optogenetic tools[14,15]. Proteorhodopsins are excellent targets for the development of such molecular devices, as they naturally exhibit spectral tuning, which can be further engineered for optimal stimulation[14,16]. In terms of efficiency, GPR is characterized by a significantly faster photocycle and consequently, an increased proton translocation rate compared to BPR[17]. Similar to its intensively studied, archaeal homologue bacteriorhodopsin (BR), GPR folds into a seven-transmembrane α-helical bundle containing a covalently bound all-*trans* retinal, which is conjugated to a conserved lysine residue via a characteristic Schiff base[5]. Despite considerable efforts, detailed structural information about the proteorhodopsin family has been limited. In recent years, a solution NMR structure[18] of GPR and crystal structures of the homologue BPR[19] have been solved. However, due to the limitations of the solution NMR structure (i.e., imposed structural restraints and monomeric state), molecular insights into the GPR family have remained sparse. Two intrinsic GPR properties might have precluded the determination of a high-resolution structure thus far: (i) very short interhelical loops, which provide insufficient hydrophilic contacts for the formation of well-ordered protein crystals, and (ii) ambiguity about the oligomeric state, which prevented the preparation of a homogeneous protein sample.

In this work, we solve the structure of pentameric GPR using single-particle cryo-electron microscopy (cryo-EM), applying our recently published procedure to isolate pure and homogeneous GPR pentamers[20]. The presented cryo-EM structure provides molecular insights into functional residues involved in the proton translocation pathway and the oligomerization interface. Comparison of GPR and BPR structures, and molecular dynamics (MD) simulations of GPR unveil conformational variations that regulate the solvent access to the intra- and extracellular cavities, harbouring the primary proton donor E109[6] and the proposed proton release group E143[21]. Based on these observations, we provide a mechanism for the conformational changes allowing the hydration of the intracellular half channel, which depends on the protonation state of E109. Functional characterization of selected mutants at the entrance and exit of the proposed proton translocation pathway demonstrates the importance of the molecular organization surrounding key residues E109 and E143. Finally, we present evidence for the scaffold function of helices involved in the oligomerization interface and showcase differences in the protein dynamics depending on the oligomeric state. Together, these findings illustrate the previously observed functional importance of GPR oligomerization[22,23].

## Results and discussion

**Overall structure of pentameric GPR.** GPR was heterologously expressed and purified without purification tags, yielding highly pure and homogeneous pentamers (Fig. S1). Using single-particle cryo-EM, we have solved the structure of GPR at a resolution of 2.9 Å (PDB ID: 7B03). GPR protomers assemble into a pentamer around a central pore with a $C_5$ symmetry axis (Fig. 1A). Individual protomers do not exhibit any major differences, even in the absence of a five-fold symmetry constraint ($C_1$) imposed during refinement. Monomers are composed of a seven-transmembrane α-helical bundle (helices A–G) typical for microbial rhodopsins[24]. However, the antiparallel β-sheet observed in BR[25] and xanthorhodopsin (XR)[26] in the extra-cellular loop between helices B and C (Fig. 1B) is missing. GPR, as well as the majority of microbial rhodopsins, is characterized by rather short transmembrane helices and connecting loops, with

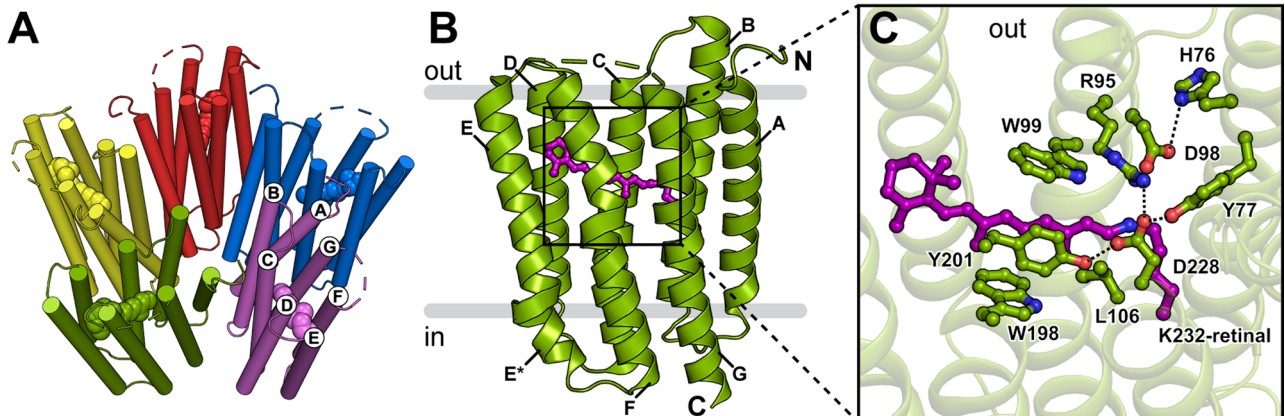

**Fig. 1 Cryo-EM structure of GPR. A** Pentameric assembly of GPR displayed from the extracellular side. Protomers (coloured individually) consist of seven-transmembrane helices (labelled A–G) and are arranged around a central pore with a $C_5$ symmetry axis. The retinal cofactors are displayed as space-filling models. **B** GPR monomer oriented in the lipid bilayer according to calculations using the PPM server[29]. The retinal chromophore is highlighted in purple. **C** Residues in the vicinity of the retinal Schiff base region, including (i) the retinal Schiff base formed with K232 (purple), (ii) the primary proton acceptor D98 in contact with H76 (distance: 3.2 Å), (iii) the counter ion D228 interacting with Y77 (2.2 Å), R95 (3.0 Å) and Y201 (2.2 Å), and (iv) W99, W198 and Y201, which contribute to the hydrophobic cavity around the retinal cofactor.

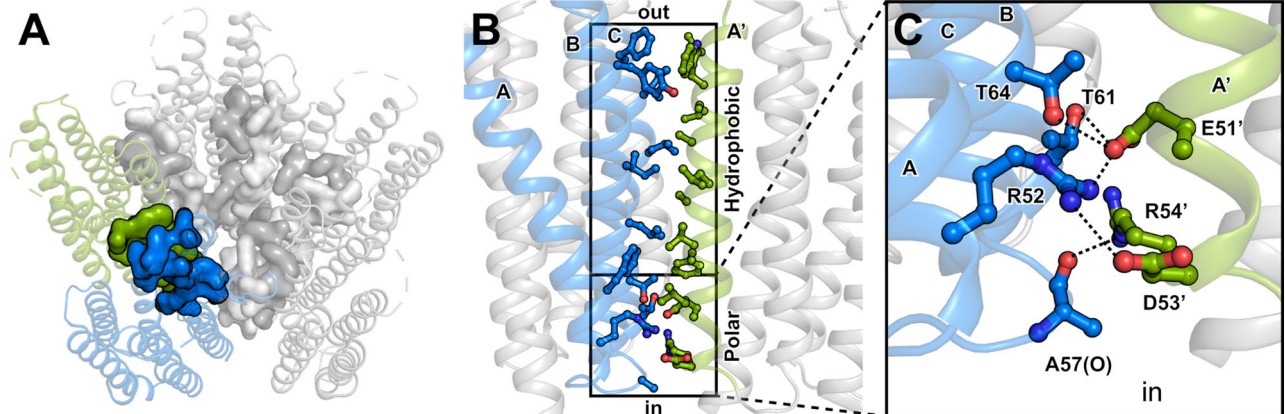

**Fig. 2 Oligomerization interface between GPR protomers. A** Pentameric GPR interacts via five identical oligomerization interfaces (surface representations) with asymmetric contributions from neighbouring protomers (green and blue). **B** The contact is stabilized by residues on helices A–C of the first (blue) and helix A′ of the second protomer (green). Main contributions to the interface stabilization include a large patch of hydrophobic interactions and a cluster of polar interactions on the intracellular side. **C** Hydrogen bonding network at the intracellular water interface comprising a number of hydrogen bonds and salt bridges between the two protomers (blue and green).

several helices barely protruding from the lipid bilayer. Most interhelical loops are formed by just a few amino acid residues. The longest loop connecting helices F and G is largely unresolved in the cryo-EM structure (indicated by dashes in Fig. 1A, B), presumably due to high flexibility.

Several helices exhibit special features, resulting from distortions of the perfect α-helical geometry (Fig. 1B and Fig. S2). Firstly, a kink is found in the middle of helix C around P104 causing it to slightly change direction. Secondly, a cytoplasmic extension of helix E (E* in Fig. 1B) is formed by residues E171-N177 following a larger turn introduced by G170, confirming previous NMR studies on the secondary structure of the E-F loop[27,28]. Notably, the start of helix E* coincides with the interface between the lipid bilayer and the aqueous environment according to calculations using the Positioning of Proteins in Membrane (PPM) server[29]. The distinct E-F loop structure has been shown to be important for the photocycle dynamics of GPR and may even modulate the local hydration dynamics, which could be relevant for the cytoplasmic proton uptake[28,30]. Despite its distance to the retinal-binding pocket, the E-F loop mutation A179R results in a significant shift of the chromophore absorption maximum and the pKa of the primary proton acceptor[31]. In the cryo-EM structure, the side chain of A179 faces the centre of the protein. Substitution with a bulky residue, such as an arginine, would thus significantly distort the local structure and result in partial unfolding of helix E*. This observation supports a proposed model for the structural consequences of the A179R mutation, which provides an interaction pathway from the distorted E-F loop to the retinal-binding pocket and explains the significant change in the pKa of the primary proton donor and the absorption maximum[30]. Lastly, π-helix segments are observed in the middle of helices F (I193-W198) and G (V230-F235). The resulting π-bulge in helix F is compensated by a 3₁₀-helix segment (W198-Y201), thus preserving the register and direction of the helix. Helical perturbations are considered to serve as hinges, which facilitate conformational changes for protein function, e.g., in substrate translocation cycles of α-helical membrane transport proteins[32,33]. The presence of these structural features points toward a conformational flexibility, potentially important for the translocation of protons in the photocycle of GPR.

**Structure of the retinal Schiff base region.** The central Schiff base is formed by covalent attachment of the retinal cofactor to

the primary amine of K232 (Fig. 1C). The well-defined cryo-EM density resolves the retinal in the all-*trans* conformation (Fig. S2), indicating a ground state GPR structure. The general spatial arrangement and interactions of the residues surrounding the Schiff base correspond to those in most closely related microbial rhodopsins[19,25,26,34,35]. Two negatively charged residues, in this case D98 and D228, are typically encountered on the extracellular side of the Schiff base, stabilizing the positive charge when protonated[36] (Fig. 1C). D98 also functions as the primary proton acceptor upon retinal photoisomerization[6]. H76, a proteorhodopsin-specific histidine residue, forms a hydrogen bond with D98, which was found to significantly increase the pKa of the proton acceptor compared to BR[8]. On the other hand, D228 is stabilized by interactions with Y77, R95 and Y201. Finally, the polar environment of the Schiff base is confined by the aromatic side chains of W99, W198 and Y201. The side chain of residue 106 largely determines the spectral tuning of proteorhodopsins, with a leucine or glutamine found in GPR and BPR, respectively[37]. The proximity of L106 explains its influence on the electronic structure and in turn the absorption maximum of the retinal chromophore.

**Interactions of the oligomerization interface.** The GPR pentamer is stabilized by five identical oligomerization interfaces with a surface of about 820 Å² according to PISA server calculation[38], with asymmetric contributions from two adjacent monomers (Fig. 2A). Interactions are mainly mediated by residues on helices A–C of the first and helix A′ of the second protomer (Fig. 2B). The majority of side-chain interactions create a large hydrophobic patch in the lipid bilayer region with an additional polar network near the intracellular solvent interface. Specifically, this comprises a cluster of hydrogen bonds and salt bridges involving residues R52, T61, T64 and the carbonyl oxygen of A57 on the first protomer, and E51′, D53′ and R54′ on the second protomer (Fig. 2C). R52 forms a salt bridge with E51′ and D53′, while E51′ also interacts via hydrogen bonds with T61 and T64. Lastly, R54′ forms a hydrogen bond to the carbonyl oxygen of A57. Disturbing these interactions by site-directed mutagenesis has been demonstrated to affect the oligomeric assembly of GPR[39].

**Comparison of GPR and BPR structures.** GPR is most closely related to the two BPR versions *Hot75*BPR and *Med12*BPR, with amino acid sequence identities of about 80% and 56%,

respectively (Fig. S3A). The identity to other microbial proton-pumping rhodopsins of known structure is only between 22–36% (Fig. S3A). In the following, the presented cryo-EM structure of pentameric GPR (PDB ID: 7B03) is compared to the monomeric solution NMR structure of GPR (PDB ID: 2L6X) and the pentameric crystal structure of the *Hot75*BPR D97N mutant (PDB ID: 4KLY; referred to as BPR from here on if not specified). For simplicity, all residue numbers refer to the corresponding positions in the presented GPR structure (UniProt accession number Q6J4G7) due to the following variations: (i) the GPR variant used in the NMR experiments (UniProt accession number Q9F7P4) is missing a glycine at position 2, resulting in a register shift of −1, and (ii) *Hot75*BPR (UniProt accession number Q9AFF7) contains an additional glycine after M211 resulting in a register shift of +1 thereafter (Fig. S3B).

The structural alignment between pentameric GPR and BPR reveals a relatively high conservation, with a root mean square deviation (RMSD) of 1.9 Å for the pentamer and an average of 1.6 ± 0.1 Å for the protomers (BPR protomers are not identical). This reflects the similarity of the overall structures, specifically the positions and lengths of the transmembrane helices and loop regions (Fig. S4A). Conversely, important deviations are observed when comparing the GPR monomer to the ensemble of NMR structures, with RMSDs ranging from 4.0 to 5.3 Å (Fig. S4B). The majority of functionally important residues between GPR and BPR are well aligned, but interestingly, different rotamer conformations are found for the primary proton donor E109 (Fig. S4C). The structure of the Schiff base region in the *Hot75*BPR D97N mutant does not seem to be significantly affected compared to GPR or wild-type *Med12*BPR (PDB ID: 4JQ6), even though the mutation of the primary proton acceptor has been shown to result in slower photocycle dynamics[40].

As indicated by larger RMSDs, significant variations in the overall structure and positions of key residues are observed when comparing the GPR cryo-EM and NMR structures (Fig. S4B, D). Striking differences include the orientation of side chains R95, E109 and E143, which are far from their expected positions in the majority of NMR models (Fig. S4E). The counterion R95 is, on average over all models, almost 9 Å away from D228, compared to 3 Å in the cryo-EM structure, thus precluding any interaction. In the vast majority of NMR models, the primary proton donor E109 is found protruding into the membrane environment, which would be energetically unfavourable and would impede transfer of protons to the Schiff base region. Similarly, the proposed proton release group E143, which is usually involved in a hydrogen bond network, protrudes into the bulk solvent without possible interaction partners. Furthermore, in the cryo-EM structure Y201 is oriented parallel to the retinal and stabilizes the counter ion D228 by formation of a hydrogen bond. In the NMR structures, it is oriented almost perpendicular to the retinal and is on average about 9 Å from D228. Finally, the primary proton acceptor D98, which usually is in close proximity to the retinal Schiff base (about 4 Å in GPR and BPR), was found at a distance of at least 5 Å.

While the NMR structures were solved from monomeric GPR, the cryo-EM structure describes the complete pentameric state, enabling analysis of the oligomerization interface (Fig. 2). Similarly, the BPR structure was also solved as a pentamer, allowing a direct comparison (Fig. S5). Almost all interactions between GPR protomers are conserved in the BPR structure (Fig. S5A, B), with the exception of a glutamine instead of an arginine at position 54 in BPR. Additional hydrogen bonds between the N-terminus of one, and helices B–C of an adjacent protomer, provide further stabilization of the oligomerization interface in BPR (Fig. S5C). The corresponding residues could not be built in the GPR model due to a lack of experimental density.

Another difference concerns a functional interaction between H76 and W35 residues of neighbouring BPR protomers[19], which is not formed in the GPR structure (Fig. S5C). Hydrogen bonding with W35 is expected to stabilize H76 in an optimal conformation for the interaction with the primary proton acceptor D98 (Fig. 1C). In GPR the tryptophan side chain is flipped towards the extracellular side relative to its position in BPR, impeding the formation of a hydrogen bond to H76. The same configuration is present in one of the *Med12*BPR (PDB ID: 4JQ6) protomer pairs. These observations and the fact that both W35 rotamer conformations were sampled during MD simulations of GPR (Fig. S6) suggest that the interaction with H76 can change dynamically and might be part of the photocycle. Notably, the W35 conformation capable of interacting with H76 is exclusively observed in the pentamer and not the monomer, due to the stabilization of additional rotamers by the oligomerization interface (Fig. S6). The functional flexibility within the W35-H76-D98 triad is further corroborated by alternating H76 tautomers observed using highly sensitive NMR measurements[9].

**Structural comparison to other microbial rhodopsins.** The most notable structural difference of proteorhodopsins compared to many microbial rhodopsins, including BR and XR, is the absence of the antiparallel β-sheet in the B–C loop, which is only a few amino acids long in GPR and BPR. Small additional differences are observed in the relative orientation and length of the transmembrane helices as well as the position of amino acid side chains in the retinal-binding pocket, all of which has been extensively described for BPR[19]. Furthermore, the proton release group in BR is comprised of two glutamates, which have been proposed to be replaced by a single glutamate in XR[26] and in both proteorhodopsin variants (E143 in GPR)[19,21]. The structures of two related light-driven proton pumps, *Exiguobacterium sibiricum* rhodopsin (ESR, PDB ID: 4HYJ)[34] and *Gloeobacter violaceus* rhodopsin (GR, PDB ID: 6NWD)[35] have been solved recently. ESR lacks the antiparallel β-sheet, similar to GPR, while GR exhibits a similar B–C loop structure as XR, with the β-sheet facing outward. Furthermore, GPR and ESR lack the 3-omega motif found in GR and other XR-like proteins[35]. The structure and identity of functional residues are almost identical in the three proteins (Fig. S7A). However, ESR contains a lysine residue instead of a glutamate as its primary proton donor and exhibits a different H57 (H76 in GPR) rotamer conformation, oriented more toward the retinal Schiff base.

A recent study has assessed the oligomeric states of a range of microbial rhodopsins from different phylogenetic branches[41]. Notably, the majority of eubacterial rhodopsins seem to assemble into pentamers. However, besides GPR and BPR, the only other structure that has been solved in a pentameric state is KR2 (PDB ID: 6REW)[42]. The KR2 oligomerization interface exhibits significantly extended polar interactions compared to GPR, which in addition to helices A–C also include helix D and the B–C loop (Fig. S7B). Many interactions are mediated by water molecules as well as a sodium ion, neither of which could be resolved in the GPR structure.

**Solvent access to the proton release group and primary proton donor.** A large hydrophilic cavity is observed in the GPR structure at the extracellular surface (Fig. 3A), coinciding with the location of the proposed proton release group E143[21]. In contrast, no solvent accessibility is observed on the intracellular side of the proton translocation pathway, which is required for reprotonation of the primary proton donor E109 during the photocycle. The extracellular cavity (Fig. 3B) provides access for the bulk solvent to E143 and R95, which is also implicated in the

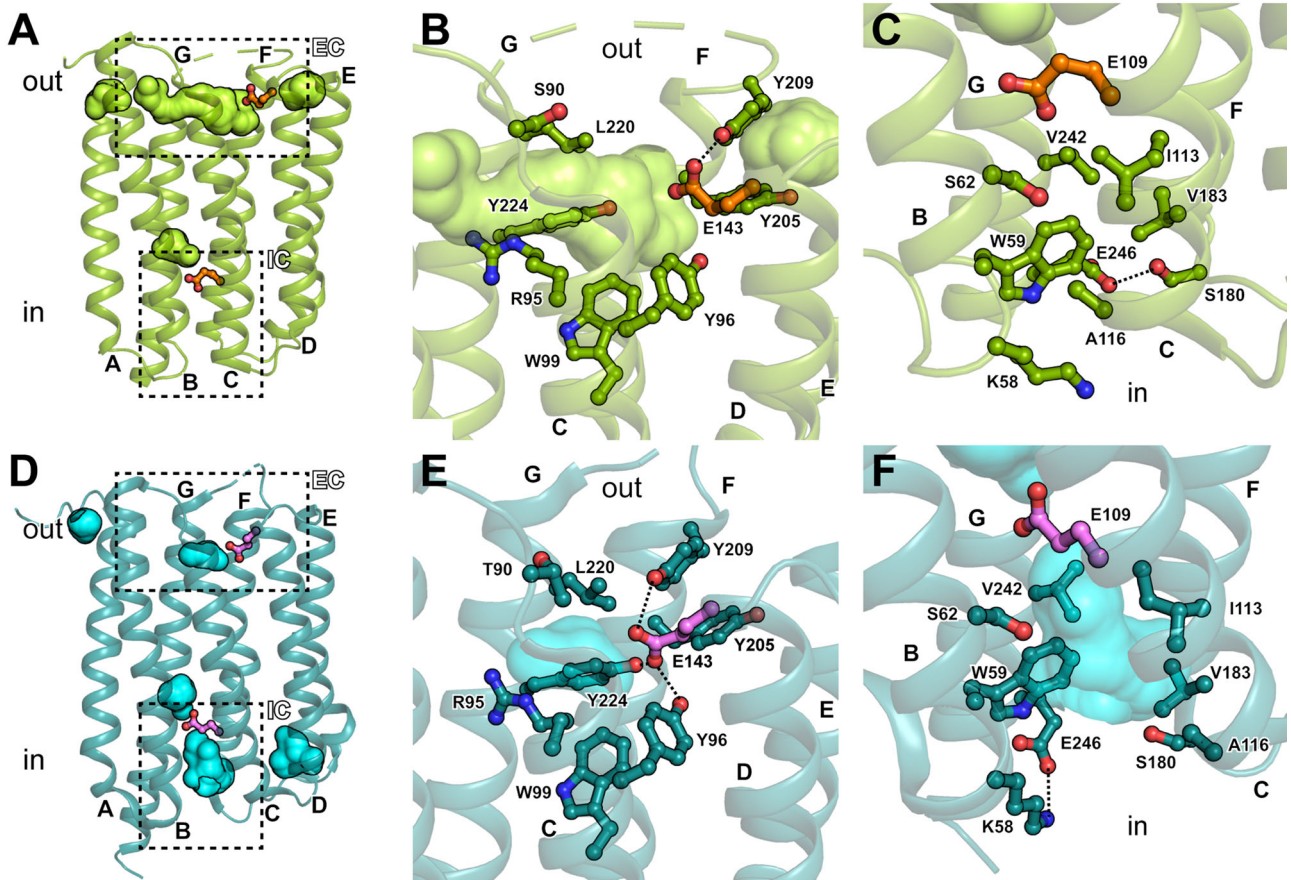

**Fig. 3 Intra- and extracellular cavities in GPR (green) and BPR (turquoise). A** GPR exhibits an outward-open state with extensive accessibility to the extracellular cavity (EC), and a completely blocked-off intracellular cavity (IC). EC and IC regions (dashed boxes) with the proposed proton release group E143 in the EC and the primary proton donor E109 in the IC are indicated in orange. Residues lining the open EC (**B**) and the closed IC (**C**) in GPR are displayed. **D** BPR exhibits an inward-open state with solvent accessible IC and obstructed EC. EC and IC regions (dashed boxes) with E143 in the EC and E109 in the IC are indicated in pink. Residues lining the closed EC (**E**) and the open IC (**F**) in BPR are shown.

extracellular proton pathway[25]. The cavity is lined by hydrophobic residues including Y96, W99, Y205, Y209, L220 and Y224. At the same time, Y209 stabilizes and possibly orients E143 through a hydrogen bond. On the intracellular side, access to E109 is completely blocked by a number of residues, mainly on helices B, C, F and G (Fig. 3C). The hydrogen bond between E246 and S180 may stabilize this closed conformation.

An inverse accessibility is observed in all BPR protomers, where the extracellular cavity and E143 are cut off from the bulk solvent, while the intracellular cavity opens up toward E109 (Fig. 3D). In addition, different rotamer conformations are observed for E109 in GPR and BPR (Fig. S4C), which may facilitate the uptake of a proton from the cytoplasm or its transfer toward the Schiff base (Fig. 3C, F). Comparing the conformations exhibited by GPR and BPR provides a model for the rearrangements required to allow access to the intra- and extracellular parts of the proton translocation pathway. Through a collective shift in helices D–G, the entrance to the extracellular cavity is obstructed. Specifically, side chains of residues Y205, Y209, L220 and Y224 are moved towards the opening of the extracellular cavity compared to GPR (Fig. 3E). While in GPR only Y209 is in hydrogen bonding distance to E143, in BPR it forms an extensive network to Y96, Y209 and Y224, and is buried inside the cavity. The increased solvent accessibility of the intracellular cavity in BPR compared to GPR results mainly from a larger interhelical distance between helices B and C (Fig. 3C, F). The open conformation in BPR favours formation of a salt bridge between

E246 and K58, replacing the interaction with S180 in GPR. This particular glutamate and its interactions seem to be partially conserved outside the proteorhodopsin family. It is also found in XR[26], in which it interacts with the conserved S180 (similar to GPR), and in GR[35], in which the glutamate is substituted by an aspartate, that interacts with the conserved K58 (similar to BPR). The conservation of this interaction suggests an involvement in the proton translocation cycle, potentially by stabilizing intermediate conformational states.

**Functional residues at the entrance and exit of the proton translocation pathway.** The interactions observed between residues in the intra- and extracellular cavities appear to be critical for proton transport by GPR due to their close proximity to the entrance and exit of the proton translocation pathway. Specifically, they might be involved in regulating solvent access for proton uptake and release. To evaluate their functional importance, we have assessed the effects of site-directed mutations on the proton pumping activity of GPR. In particular, we have focused on the following largely uncharacterized mutations: K58A, W59A, S180A and E246A on the intracellular side (Fig. 4A) and Y96F, E143A, Y209F and Y224F on the extracellular side (Fig. 4B). Removal of the proposed proton release group E143 or the interacting hydroxyl groups of Y96 and Y224 (as seen in BPR, Fig. 3E) reduced GPR photoactivity by about 50%. The diminished proton pumping activity of Y96F also confirms previous findings[43], whereas the effect of Y209F was

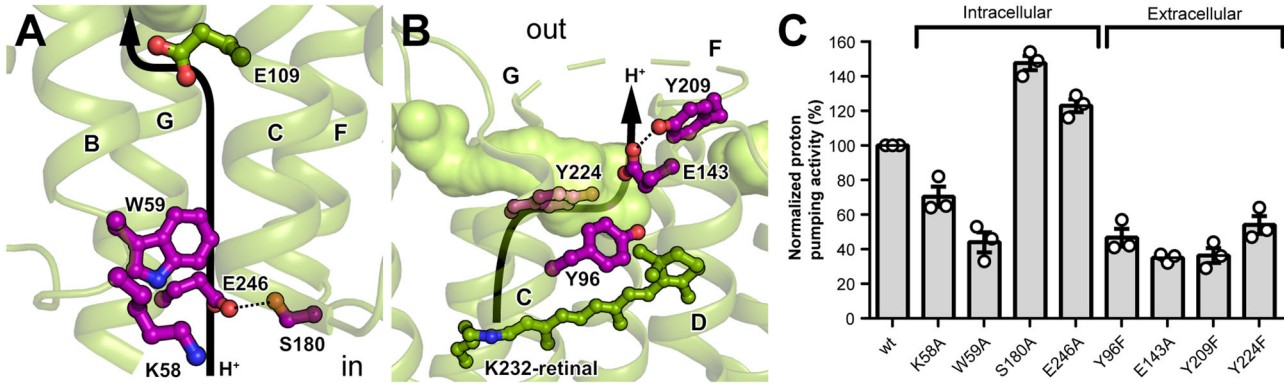

**Fig. 4 Functional residues at the entrance and exit of the proton translocation pathway in GPR. A** Closed intracellular cavity containing the primary proton donor E109 and **B** open extracellular cavity harbouring the proposed proton release group E143. Mutated residues surrounding the entrance and exit of the proposed proton translocation pathway (indicated with black arrow) are highlighted in purple. **C** Proton pumping activities of wild-type (wt) GPR and selected mutants. Each bar represents the mean of three independent experiments (shown as circles) performed in triplicate with indicated standard error of the mean (SEM). Values from individual experiments were normalized to the activity of wt GPR and corrected for their relative expression levels (Table S1). Source data are provided as a Source data file.

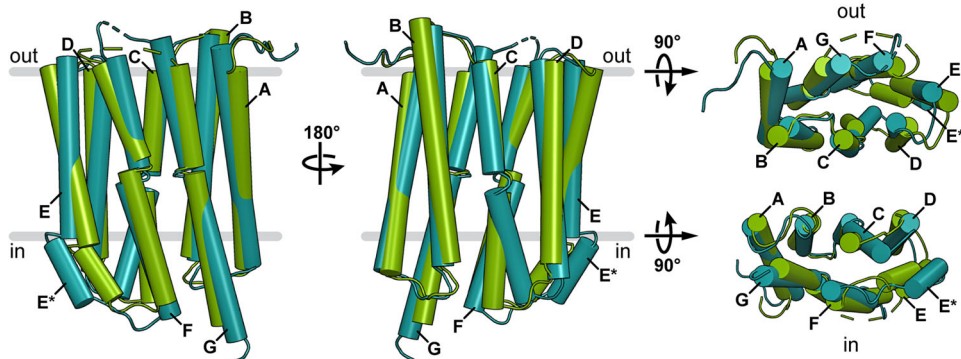

**Fig. 5 Conformational differences between the GPR outward-open (green) and BPR inward-open (turquoise) states.** Relative orientations of the different points of view are indicated with rotation arrows. In the side views the location of the membrane is highlighted according to PPM server calculation[29], with the extracellular (out) and intracellular side (in) indicated. Helices are labelled A–G and hinge regions are illustrated by breaks in the helices.

slightly more pronounced (Fig. 4C). Furthermore, we observed diminished proton pumping activities for K58A and W59A, which are located at the entrance of the proton translocation pathway (Fig. 4A, C). The reduction in photoactivity upon substitution of these residues supports their proposed importance in the proton transport by GPR. Interestingly, the proton pumping efficiency markedly increased when the interaction between S180 and E246 (Fig. 4A) was removed. This suggests that the interaction is slowing down the photocycle by stabilizing an intermediate state, which could have evolved to regulate GPR activity.

**Conformations regulating the intra- and extracellular solvent access**. The potential helix movements involved in the conformational change between the outward-open and the inward-open states are inferred from the superposition of GPR and BPR (Fig. 5) and are illustrated by a morph between the two structures (Movie S1). The closing of the extracellular cavity is mainly mediated by movement of helices D–G towards the protein centre. The major contribution to sealing the intracellular cavity is provided by the C-terminal segment of helix C and helices E* and G (Fig. 5). In contrast, helices A, B and the N-terminal part of helix C seem to serve as scaffold against which the other helices can pivot. This is supported by the fact that the same parts of the protein are involved in forming the oligomerization interface (Fig. 2 and Fig. S8A, B). When comparing MD simulations of

monomeric and pentameric GPR, we observed higher Cα root mean square fluctuations (RMSF) for residues in the monomer, indicating higher flexibility (Fig. S8C). The increased rigidity of the pentamer likely originates from the stabilization provided by the oligomerization interface, which involves helices A–C. The scaffold function of these helices bears resemblance to the fixed helices in the rocking-bundle mechanism of many membrane transport proteins[44]. Furthermore, the reduced overall flexibility in the pentamer might ensure optimal orientations of amino acid side chains and water molecules involved in the proton translocation and could indicate a regulatory function for the oligomeric state. Importantly, the scaffold function of helices A–C and the increased rigidity of the pentamer substantiate the functional relevance of GPR oligomerization, which has been observed previously. Specifically, GPR oligomerization lowers the pKa of the primary proton acceptor D98 and prolongs the decay time of the M intermediate compared to the monomer[22,23]. The drastic change in the pKa of D98 can probably be attributed to the missing interprotomer interaction between W35 and H76 (Fig. S5C), while the increased rigidity of the pentamer (Fig. S8C) could contribute to the slower photocycle kinetics.

The largest movements of helix segments observed when comparing the GPR and BPR structures correspond to regions endowed with high flexibility by the described helix disturbances that act as potential hinges (illustrated by broken helices in

**Table 1 Set-up parameters for MD simulations with GPR.**

| Oligomeric state | Protonated glutamates | System size [Å] | Number of replica | Simulation time of each replica [μs] |
|---|---|---|---|---|
| monomer | E109/E143 | 72 × 72 × 99 | 3 | 0.5 |
| pentamer | E109/E143 | 140 × 140 × 99 | 2 | 1.6 |
| pentamer | – | 140 × 140 × 99 | 2 | 1.6 |
| pentamer | E109 | 140 × 140 × 99 | 2 | 0.75 |
| pentamer | E143 | 140 × 140 × 99 | 2 | 0.75 |

Fig. 5). These allow the N- and C-terminal parts of the helices to move independently to some degree. Based on these structural differences, we propose a change in conformation that allows the hydration of the intra- or extracellular cavity, which could be triggered by or regulate the protonation states of the primary proton donor E109 and the proposed proton release group E143 during the photocycle.

**Protonation-dependent conformational dynamics and hydration of half channels.** In order to test if the conversion between inward- and outward-open GPR conformations is thermodynamically feasible, we carried out four sets of MD simulations (Table 1), with different protonation states for E109 and E143 (i.e., both residues deprotonated, one deprotonated and the other protonated or both protonated). Two distinct protonation-dependent conformations were sampled during the simulations (Fig. S9A, B). The GPR cryo-EM structure coincides with the conformation in which either both glutamates or only E109 are protonated. It is likely, that in our structure both glutamates are protonated due to the pH at which the structure was solved, i.e., pH 7.5, and the reported pKa values of both carboxylic acids (pKa > 8.5 for E109 and > 9.0 for E143)[21,45]. The second population of conformations is sampled by GPR with both residues deprotonated, or only E143 protonated. The BPR structure coincides with the double deprotonated state while it is slightly less comparable to the E143 protonated state (Fig. S9A, B). This indicates, that the protonation state of E109 largely determines the conformation, whereas that of E143 seems to have a minor effect.

Furthermore, we have specifically investigated the opening and closing of the intra- and extracellular cavities during the MD simulations. We used the distance between two residue pairs (W59/A116 and G139/L220) as a measure for the conformational changes (Fig. S9C–E) and assessed the hydration dynamics of the respective half channels (Figs. 6 and S10). In the double protonated state, the extracellular cavity exhibits a pronounced conformational flexibility, while the motion of the intracellular side is more limited (Fig. S9C). This is also reflected by the number of water molecules observed in the respective half channels (Fig. S10). Deprotonation of E109 triggers the opening of the intracellular and slight constriction of the extracellular cavity, resulting in an inward-open conformation similar to BPR, but with a more accessible extracellular cavity. Importantly, we could observe a concurrent influx of water into the intracellular half channel (Figs. 6 and S10), which would facilitate reprotonation of E109 at the end of the photocycle. A similar behaviour has been described for BPR[46] and BR[47]. This conformational change is thermodynamically accessible under ambient conditions and does not require retinal isomerization, but depends mainly on the protonation state of E109. Furthermore, it is conceivable that the difference in E109 rotamer conformation (Fig. S4C) could result from different protonation states, i.e., deprotonated in BPR and protonated in GPR. We have also assessed the frequency of the functionally important interaction between S180 and E246 (Fig. 4) during the simulations. Hydrogen bonding was observed more

frequently when E109 was protonated (about 83% of the time) compared to when it was deprotonated (about 50% of the time). This supports the hypothesis that the interaction might preferably stabilize a state with closed intracellular half channel, which might slow down the transport cycle.

Despite the seemingly small effect of deprotonating E143 on the overall conformation (Fig. S9), the extracellular half channel experiences a significant increase in hydration (Figs. 6 and S10). This indicates a similar gate keeper function for E143 as demonstrated for E109 and would further support its assignment as the proton release group of GPR. Interestingly, the increased water density in the extracellular half channel is mostly localized to the proximity of E143 (Fig. 6).

In summary, our results provide molecular insights into the structure, dynamics and function of the proteorhodopsin family. In particular, the pentameric GPR cryo-EM structure resolves important functional residues of the proton translocation pathway and interactions stabilizing the oligomerization interface. MD simulations provide evidence for the scaffold function of helices directly involved in the protomer interface and reveal an increased rigidity for the pentamer, supporting the previously observed functional importance of GPR oligomerization[22,23]. Furthermore, conformational differences between GPR and BPR provide a potential structural model for the regulation of solvent access to the intra- and extracellular cavities, which accommodate the primary proton donor E109 and the proposed proton release group E143. Functional characterization of selected mutants demonstrates the importance of the molecular organization around E109 and E143 for the proton pumping function of GPR. MD simulations of GPR reveal that the opening and closing of the intracellular half channel and its hydration are induced by a change in the protonation state of E109, similar to findings for BPR and BR[46,47]. A comparable effect is observed for E143, where deprotonation results in an increased hydration of the extracellular half channel but without requiring significant conformational changes. This suggests that the protonation states of E109 and E143 might act as triggers for the hydration of the intra- and extracellular cavities. Changes in the protonation state of E109 are proposed to occur during the later stages of the GPR photocycle. In these models, the primary proton donor transfers its proton to the Schiff base during the M- to N-state transition and is reprotonated during the following conversion to the O-intermediate, which is the final stage before returning to the ground state. The observed closing of the intracellular half channel could thus describe part of the conformational change during the N–O transition resulting from the reprotonation of E109. Due to its status as a model protein in the field, these results will be valuable for the large scientific community employing GPR for the development of novel biophysical techniques and applied technologies.

## Methods
**Cloning and overexpression of GPR.** The wild-type GPR gene (GenBank: AY601905.1) was cloned and overexpressed in *Escherichia coli* BL21(DE3) Rosetta2 as described previously[20]. This yields native-like GPR without signal sequence and

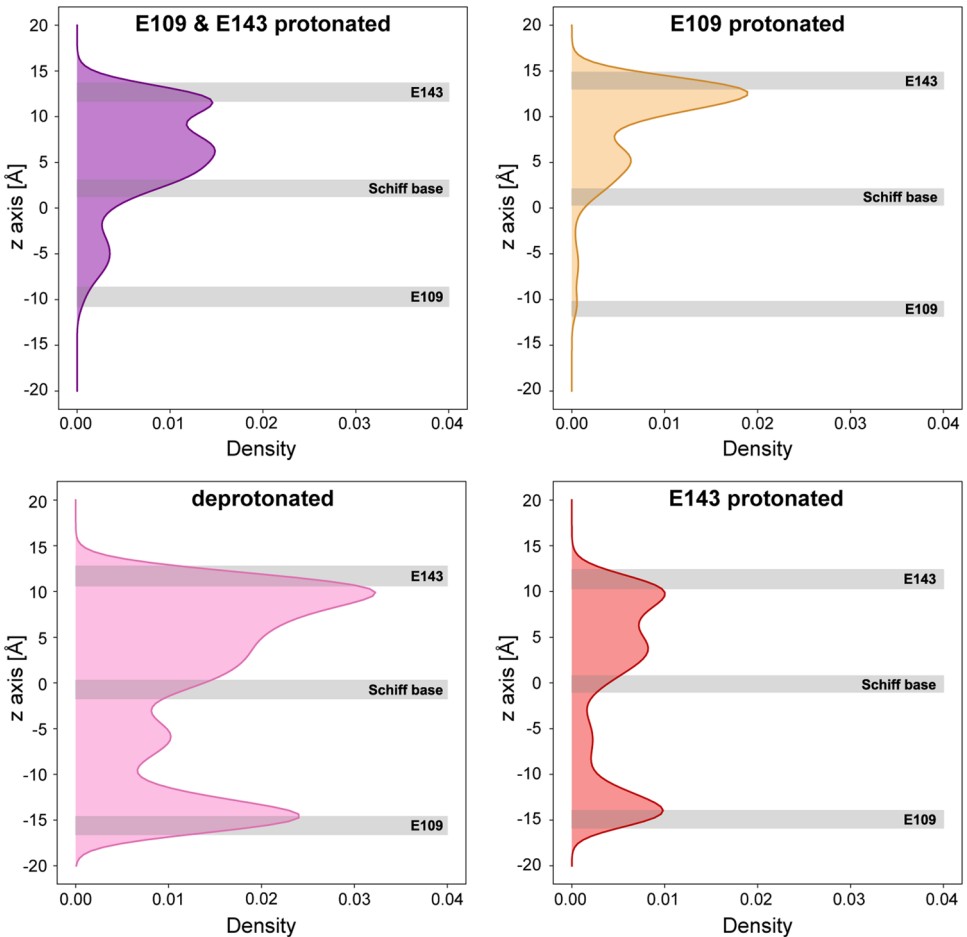

**Fig. 6 Hydration of GPR channel during the MD simulations.** The distribution of water molecules along the main axis of the GPR channel is computed using a kernel density estimate. The membrane is centred at the origin and oriented along the $z$-axis. The average positions of E109 and E143 carboxyl groups and the nitrogen of the Schiff base are shown with grey boxes. The height of the box corresponds to the standard deviation of each position.

without purification tags, corresponding to amino acid residues 19–250 with an N-terminal methionine.

**Tag-free purification of GPR.** Tag-free purification of GPR was performed essentially as described previously[20]. 5-cyclohexyl-1-pentyl-β-D-maltoside (Cymal-5, Anatrace) was substituted for 4-cyclohexyl-1-butyl-β-D-maltoside (Cymal-4, Anatrace) using final concentrations of 3% (w/v) for membrane solubilization and 0.8% (w/v) for all remaining steps. A detailed description of the experimental procedure and biochemical characterization of the purified protein can be found in the Supplementary Information (subchapter Tag-free Purification of GPR and Fig. S1).

**Cryo-EM grid preparation and data collection.** Purified GPR (2.5 µL at 3.5 mg/mL) was applied to Quantifoil R1.2/1.3 300 mesh copper holey-carbon grids that were glow discharged for 30 s at 10 mA and 0.25 mbar. Samples were blotted for 5 s and vitrified in liquid ethane using a Vitrobot Mark IV (ThermoFisher) at 4 °C and 100% humidity. Data were acquired using SerialEM[48] on a Titan Krios G3 (ThermoFisher) operated at 300 kV and equipped with a Quantum-K3 direct electron detector (Gatan). Micrograph movies were recorded in counting mode at a magnification of ×130,000 (corresponding to a calibrated pixel size of 0.645 Å) and a defocus range of −0.9 to −1.9 µm. A total of 20,307 movies were collected, each comprising 40 frames, with a total dose of 54.23 e⁻/Å².

**Data processing.** The single-particle cryo-EM image processing workflow is summarized in Fig. S11. Dose weighting and beam-induced motion correction of dose-fractionated and gain-corrected movies (2x binned to a pixel size of 1.29 Å) were performed using MotionCor2 (version 1.3.1)[49]. Contrast transfer function (CTF) parameters were estimated using ctffind 4.1.13[50]. From 20,307 micrographs, those with astigmatism greater than 300 Å, figure of merit smaller than 0.05 and maximum CTF resolution worse than 5 Å were rejected, resulting in 17,213 images. Initial particles to generate templates for autopicking were picked using a Laplacian-of-Gaussian filter in Relion 3.1[51,52]. Using selected 2D classes as templates, a total of 3,645,628 particles were picked from 17,213 images. Particles were

sorted by two rounds of 2D classification to remove low-quality particles and contaminations. 2,005,745 particles were further selected by two rounds of 3D classification using an imposed $C_5$ symmetry. 717,107 particles from classes showing high-resolution features were selected and further processed with per-particle CTF refinement and Bayesian Polishing[52]. The polished particles were imported in cryoSPARC[53] and used to generate an ab initio 3D model that was further refined by homogeneous and subsequent non-uniform refinement. The resulting map was subjected to the PHENIX[54,55] density modification tool Resolve Cryo-EM[56]. The final protein model was achieved by several iterations of manual model building in Coot[57], real-space refinement in PHENIX[54] and structure validation using MolProbity[58]. Data collection, model refinement and validation statistics are summarized in Table S2 and analysis of the final cryo-EM density map is shown in Fig. S12. The cryo-EM map and the protein coordinates were deposited in the Electron Microscopy Data Bank (EMD-11955) and Protein Data Bank (PDB-ID: 7B03), respectively. All volume and structural representations were prepared using Chimera v1.12[59] or PyMol v2.3 (The PyMol Molecular Graphics System, Schrödinger).

**Photoactivity of GPR mutants.** GPR mutants K58A, W59A, Y96F, E143A, S180A, Y209F, Y224F and E246A were prepared using the QuikChange Lightning Multi Site-Directed Mutagenesis Kit (Agilent). The variants were expressed in *E. coli* identically to the wild-type protein and as described previously[20]. Expression levels of the different GPR mutants were quantified (Table S1) and later applied to correct the proton pumping activity (Fig. 4C). To this end, cells were thawed, lysed by sonication (Branson 450 Digital Sonifier) and membranes isolated as described for the wild-type protein (see Supplementary Information). Small aliquots of GPR containing membranes were solubilized in 3% (w/v) Cymal-4 and unsolubilized material was removed by ultracentrifugation at 100,000 × *g* and 4 °C for 20 min. The optical density was measured at the retinal absorption maximum of the individual variants (Table S1) using a UV-1600PC spectrophotometer and was used to calculate the GPR concentration. Cells for the functional assay were washed once after expression in 150 mM NaCl (adjusted to pH 7.5 with NaOH) and resuspended to an OD600 of 40. Aliquots of 0.8 mL were supplemented with 0.8 mL

50% (v/v) glycerol, flash-frozen in liquid nitrogen and stored at −80 °C until used. The proton pumping activity of GPR mutants expressed in *E. coli* was assessed using an established photoactivity assay[12,60]. Briefly, cells were thawed and washed twice before resuspending in 150 mM NaCl (adjusted to pH 7.5 with NaOH). pH changes upon repeated illumination were measured using a micro pH-electrode (InLab Micro Pro, Mettler Toledo). Four consecutive cycles of darkness and illumination (8 min each) were recorded. Maximal differences of the proton concentration before and after illumination were averaged and normalized to the wild-type protein. Lastly, activities were corrected for the different expression levels of the GPR variants (Table S1).

**Molecular dynamics simulations.** Molecular dynamics simulations were carried out in order to characterize the effect of oligomerization and protonation of E109 and E143 on the dynamics of GPR. The cryo-EM structure of GPR (PDB ID: 7B03) was used as initial model. One monomer and the entire pentamer were each embedded into a lipid bilayer mimicking a bacterial membrane[61] composed of a 3:1 mixture of palmitoyl oleoyl phosphatidylethanolamine (POPE) and palmitoyl oleoyl phosphatidylglycerol (POPG). Each system was then solvated by adding a 20 Å water layer on both sides of the membrane and neutralized with 150 mM NaCl. These systems were initially assembled using the CHARMM-GUI webserver[62]. The retinal topology[63] and the glutamate protonation states were built with the psfgen plugin of VMD[64], using the force field parameters from a previous BPR study[65]. In total, five systems were assembled with different protonation states for E109 and E143 (Table 1). The protonation states of the remaining ionisable residues were assigned according to previous MD work on BPR[65].

The simulations were performed with the CHARMM36 force field[66], including CMAP corrections for the protein. The water molecules were described with the TIP3P water[67] parameterization. The simulations were carried out with OPENMM molecular engine[68] following the minimization and equilibration protocols provided by CHARMM-GUI. The cutoff for non-bonded interactions was set to 12 Å with a switching distance at 10 Å. The periodic electrostatic interactions were computed using particle-mesh Ewald (PME) summation with a grid spacing smaller than 1 Å. Constant temperature of 300 K was imposed by Langevin dynamics with a damping coefficient of 1.0 ps. Constant pressure of 1 atm was maintained with Monte Carlo barostat[69]. The hydrogen mass repartitioning scheme was used to achieve a 4 fs time-step[70]. The number of replica and simulation timescales of each system are provided in Table 1. The trajectories were analysed with VMD and an in-house script implemented in tcl. The principal component analysis was computed using ProDy[71].

**Reporting summary.** Further information on research design is available in the Nature Research Reporting Summary linked to this article.

## Data availability

The cryo-EM map was deposited in the Electron Microscopy Data Bank (EMDB) under accession code EMD-11955 and the protein coordinates were deposited in the Protein Data Bank (PDB) with the accession code 7B03. Source data are provided with this paper.

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

## Acknowledgements

Sample pre-screening was performed on equipment supported by the Microscopy Imaging Centre (MIC), University of Bern, Switzerland. We thank the Electron Microscopy Core Facility (EMCF) at the European Molecular Biology Laboratory (EMBL) and especially Felix Weiss for their excellent support. This work was supported by the University of Bern, the National Centre of Competence in Research (NCCR) Molecular Systems Engineering, the NCCR TransCure and the Swiss National Science Foundation (SNSF; grants CRSII5_183481 and CRSK-3_190705).

## Author contributions

S.H. and D.F. conceived and designed the research. S.H., D.K., T.L. and Z.U. performed the experiments. S.H., D.K., T.L. and D.F. analysed the data. S.H., T.L. and D.F. wrote the manuscript. All authors contributed to manuscript revision and approved the final version.

## Competing interests

The authors declare no competing interests.
