## [Peer Review File · Nature Communications]

REVIEWER COMMENTS

Reviewer #1 (Remarks to the Author):

In this paper, Hirschi and coworkers report a cryo-EM structure of the pentameric green-light absorbing proteorhodopsin (GPR). The resolution achieved by this method is undoubtedly significant, especially for a protein of this size in its apparent native oligomeric state. I do however have significant concerns as to the novelty of the work and the robustness of the conclusions drawn.

In mid-2020 the authors published a paper in the Journal of Structural Biology (reference 20, <https://doi.org/10.1016/j.yjsbx.2020.100024>), in which they reported success in obtaining both an X-ray crystal structure (4.4 Å resolution) and a cryo-EM structure (15 Å resolution) of green-light absorbing proteorhodopsin (GPR). Unfortunately this not only means that the paper submitted now is not as novel as it might appear at first glance, but also that the basic characterisation (purity, UV-vis and CD spectra etc) of the protein used in this study has already been published. The editor may therefore wish to consider whether the paper is sufficiently novel for Nature Communications.

The authors report significant differences between their GPR structure and a D97N mutant of a pentameric blue-light absorbing proteorhodopsin (BPR), which has a delayed photocycle relative to the wild-type protein. The authors identify GPR as having an 'outward open conformation' in contrast to the BPR structure, which exhibits an 'outward open conformation'. Although both structures include retinal in the all-trans conformation (and would therefore represent the ground state), the authors skate over this difference, by suggesting that the "relaxed ground state of proteorhodopsin could be characterized by alternating conformations, resulting in the selective hydration of the intra- and extracellular cavities". This explanation would appear to be a little unsatisfactory, since it would imply that there is a small energy barrier (relative to kT) for the conversion between forms and, if that were to be the case, then the protein samples prepared for cryo-EM would include a mixed population of the two states. They also identify differences in internal hydrogen bonding between the GPR and BPR structures and these bonds which would appear to be too strong to allow significant conversion between two states at physiological temperatures. This proposal requires further evidence before it can be confidently advanced and I would suggest to the authors that they present computational calculations (which might include simulation of a GPR D96N mutant) in order to confirm whether conversion between the two states is possible without retinal isomerisation. They should also make a detailed comment on the thermodynamics of the mechanism of conversion between the two states.

A great strength of this paper is that the authors present a structure for pentameric GPR. They claim that their structure "substantiates the functional relevance of GPR oligomerization", however they do not explain this further. An oligomerisation interface (involving helices A, B and the N-terminal part of C) is identified, however these helices show only minimal changes in orientation between the 'inward open' and 'outward open' conformations. The functional relevance of the pentamer is therefore unclear, as it would be that the same range of movement of individual helices is permitted in both the monomeric and pentameric forms. There is no suggestion (or obvious mechanism) for cooperative behavior between protomers. The authors should make clear what they mean by the "functional relevance of GPR oligomerization", explain what conformational changes are restricted/promoted in monomeric vs pentameric protein and identify potential molecular mechanisms which might make ion transport more efficient in the pentamer.

Minor suggestions to aid clarity:

Abstract

- "prototype" is the wrong word to use - "archetype" would be better, but consider rephrasing instead so that the word is not used.

- the phrase "inverse solvent accessibility" is unclear

Introduction

Page 3

- Line 3 - change "photochemical reaction centers, using a chlorophyll cofactor" to "chlorophyll-containing photochemical reaction centers"
- Line 12 - delete "respectively"
- Line 13 - change "and characterized as a light-driven proton pump" to "and was characterized as a light-driven proton pump"
- Line 14 - "prototype" is the wrong word to use here
- Line 16 - change "bionanotechnological applications" to "applications in bionanotechnology"
- Line 18 - change "focus of efforts to implement them as optogenetic tools" to "basis for the engineering of optogenetic tools"
- Line 21 - change "and a consequently increased proton translocation rate" to "and consequently, an increased proton translocation rate"

Page 4

- Line 2 - change "containing a covalently attached all-trans retinal, which forms the characteristic Schiff base with a conserved lysine residue" to "containing a covalently bound all-trans retinal, which is conjugated to a conserved lysine residue via a Schiff base"
- Line 6 - change "crystal structures of the homolog BPR were solved" to "crystal structures of the homolog BPR have been solved"
- Line 8 - change "molecular insight in the GPR family has remained sparse" to "molecular insights into the GPR family have remained sparse"
- Line 11 - change "ambiguity about the oligomeric state prevented the preparation of a homogeneous protein sample" to "ambiguity about the oligomeric state has prevented the preparation of a homogeneous protein sample"

Results

Page 4

- Line 3 - delete "thus enabling structural characterization as similar to the native version as possible"

Page 5

- Line 3-14 - delete from "We compare the GPR structure..." to "functional importance of GPR oligomerization." This paragraph is too long and too introductory.

Page 6

- Line 2 - replace "comprises" with "has"
- Line 12 - the abbreviation "OPM" should be explained

Page 7

- Fig 1 caption - authors should give the distances between the following residue pairs (either in the caption or as a table in the Supplementary Info): D98 and H76, D98 and D228, D228 and Y201, D228 and Y77.

Page 8

- Line 3 - replace "proteorhodopsin specific" with "proteorhodopsin-specific" (hyphen required)

Pages 10 and 11

The very long paragraph should be split into at least three smaller paragraphs

Page 12

- Line 19 - replace "which was also implicated" with "which is also implicated"
- Line 20 - replace "The

outline of the cavity is created by an accumulation of hydrophobic residues including" with "The cavity is lined by hydrophobic residues including"

Page 13

- Line 2 - replace "This includes the formation of a hydrogen bond between E246 and S180, which might contribute to stabilizing this closed conformation." with "The hydrogen bond between E246 and S180 may stabilize this closed conformation."
- Line 3 - new paragraph before "An inverse accessibility"
- Line 3 - replace "An inverse accessibility is observed in all BPR protomers, where the extracellular cavity" with "Conversely, in all BPR protomers, the extracellular cavity"
- Line 15 - new paragraph before ""An inverse accessibility"
- Line 16 - replace "favours" with "favors"
- Line 17 - delete "Interestingly"
- Line 18 - delete "also"
- Line 19 - replace "xanthorhodopsin (XR), interacting with the conserved" with xanthorhodopsin (XR), in which it interacts with the conserved"
- Line 20 - replace "Gloeobacter rhodopsin (GR), where the glutamate is substituted" with "Gloeobacter rhodopsin (GR), in which the glutamate is substituted"

Page 14

- Fig 3 caption - replace "E109 in the IC are indicated (orange)." with "E109 in the IC indicated in orange."
- Fig 3 caption - delete "state with inverse solvent accessibility compared to GPR"
- Fig 3 caption - replace "E109 in the IC are indicated (pink)." with "E109 in the IC indicated in pink."

Page 15

- Line 5 - delete "the solvent accessibility"
- Line 11 - replace "seem to serve as scaffold for the described motion of the other helices." with "seem to serve as a scaffold against which the other helices may pivot."
- Line 19 - replace "hypothesize" with "propose"

Reviewer #2 (Remarks to the Author):

This manuscript reports the first medium-high-resolution structure of Green Proteorhodopsin (GPR), the prototype of the large protein family of eubacterial proteorhodopsin proton pumps. The authors did employ a spectacular array of experimental and in silico methodology to unravel electron-microscopical images to submolecular resolution. Comparison with literature data of its distant cousin blue proteorhodopsin (BPR), apparently crystallized in a different functional state, results in a very workable model for the global functional mechanism of GPR and possibly ion-pumping rhodopsins in general. Still, a more detailed discussion of the differences in protomer structure between the pentameric state and other states, reported in the literature, would also be very informative. Further, the technical approach is well outlined, properly calibrated and persevered in depth, with careful data analysis and extensive in silico data reduction. The protein structure has been publicly deposited in the Protein Data Bank. Relevant literature has been properly covered. I would recommend that the authors take the following comments and suggestions into account.

1. The authors more or less claim that the pentameric state is the native one for GPR. Of course, this has not been analyzed for the native organism and in the membrane of the heterologous host *E. coli*

several oligomeric states have been detected. A better wording would be that the pentameric state is the most prominent oligomeric state.

2. Isolation and “clean” purification of GPR is essential in this research and merely referring to a recently published paper is not sufficient. A brief method section covering the essentials of these procedures should be included in the supplementary information. It is recommended to also include a SEC profile of the purified GPR.

3. Eventually less than 20 % of the more than 3.6×10^6 particles was selected to proceed through the final processing and refinement software packages. It would be useful to put in whether the authors did find any evidence for other functional and/or oligomeric states in the rejected particles. Would it be an option to screen the rejects with a single optimized protomer structure in search for such other states ?

4. What is the advantage of using Cymal-5 over Cymal-4, and more general of using Cymal's instead of DDM ? Would the detergent selection influence the protomer interaction pattern or helix movements to a significant extent as compared to a membrane environment ?

5. According to ref. 20 a C-terminal his-tag still allows pentameric oligomerization. It would be valuable information for the rhodopsin field when any evidence can be extracted from the high-resolution structure as to whether a his-tag might still perturb the pentameric organization or helix movements.

6. It would be useful to elaborate on the differences in protomer interaction between the pentameric state and the putative hexameric model in ref. 7 (Stone et al, 2013). Likewise, the possible structural consequences of monomerization would be very interesting, also in its potential bearing on the divergence with the NMR-based structure.

7. The entire procedure evolving into the final structural model involves a very complex process of software based selection and refinement. This of course also poses some risks. For instance:

- In contrast to the BPR pentamer, all subunits of the GPR pentamer come out identical. Can the authors exclude that this is an unfortunate consequence of the very complex in silico processes.
- The resolution is, as expected, not evenly distributed over the entire protein, with about 2.9 Å in the best resolved regions. Does this include the chromophore and binding pocket residues ? For instance, the electron density profile around the chromophore (Fig. S1) could suggest a mixture of an all-trans and a 13-cis configuration !? Further, I presume that water molecules and H-bonds have not been resolved ? Nonetheless, such ambiguities will most likely not downgrade the global functional mechanism proposed in movie S1.

Reviewer #3 (Remarks to the Author):

The article by Hirschi et al., describes the cryo-EM structure of a proteorhodopsin and deciphers novel insights into the functioning of this light driven proton pump. Using the structural template, they discuss how conformational changes associated with various transmembrane helices facilitate the pentameric assembly of this protein. While the overall structural features are interesting and provide important insights, the lack of functional data to corroborate the structural findings somewhat limit the enthusiasm for this manuscript. For example, Figure 3 and related discussion deals with how various residues and their interactions may govern proton release and translocation cycle, but experimental data in the form of site-directed mutagenesis to corroborate these observations is not presented. Even if these data are not straightforward to generate, at least a more extensive effort should be taken to discuss this in the context of existing literature on other members of this family. The method section is way too brief and does not sufficiently describe the details of protein isolation and characterization.

Overall, this is an interesting manuscript that should be published after some critical revision to better connect the structural observations with mechanistic aspects.

Reviewer #1:

In this paper, Hirschi and coworkers report a cryo-EM structure of the pentameric green-light absorbing proteorhodopsin (GPR). The resolution achieved by this method is undoubtedly significant, especially for a protein of this size in its apparent native oligomeric state. I do however have significant concerns as to the novelty of the work and the robustness of the conclusions drawn.

In mid-2020 the authors published a paper in the Journal of Structural Biology (reference 20, <https://doi.org/10.1016/j.yjsbx.2020.100024>), in which they reported success in obtaining both an Xray crystal structure (4.4 Å resolution) and a cryo-EM structure (15 Å resolution) of green-light absorbing proteorhodopsin (GPR). Unfortunately this not only means that the paper submitted now is not as novel as it might appear at first glance, but also that the basic characterisation (purity, uv-vis and CD spectra etc) of the protein used in this study has already been published. The editor may therefore wish to consider whether the paper is sufficiently novel for Nature Communications.

Authors: The authors thank the Reviewer for appreciating the achievements of the presented manuscript. Our previous publication (Hirschi et al., Journal of Structural Biology: X 2020) focused mainly on the novel tag-free purification procedure to isolate pure and homogeneous pentameric GPR. However, we did not solve the protein structure at that time. We presented evidence for the pentameric state of GPR based on X-ray crystallography and cryo-EM data. Specifically, this included a low-resolution electron density map and low-resolution 2D classes but no atomic model.

In the presented manuscript, we have provided the first high-resolution structure of GPR. Furthermore, we have significantly improved the manuscript by implementing the excellent suggestions of all three Reviewers in the revised version. In particular, this includes performing and analyzing several sets of molecular dynamics (MD) simulations of pentameric and monomeric GPR using the solved structure, as well as functional analysis of key mutations to corroborate our mechanistic conclusions. These novel results provide evidence for i) the proposed structural rearrangements between an outward-open and inward-open conformation and, importantly, their dependence on the protonation state of the primary proton donor E109, ii) the functional relevance of GPR oligomerization by confirming the scaffold function of helices A-C and revealing an overall increased rigidity of the pentamer, and iii) the functional importance of uncharacterized residues at the entrance and exit of the proton translocation pathway.

The new results have been included in the revised version of the manuscript in form of new figures (Fig. 4, Table 1, Fig. S6 and S7, Table S1), a new subchapter of the Results section (Functional Residues at the Entrance and Exit of the Proton Translocation Pathway) and extensive elaborations on previous statements (see tracked changes).

The authors report significant differences between their GRP structure and a D97N mutant of a pentameric blue-light absorbing proteorhodopsin (BPR), which has a delayed photocycle relative to the wild-type protein. The authors identify GPR as having an 'outward open conformation' in contrast to the BPR structure, which exhibits an 'outward open conformation'.

Although both structures include retinal in the all-trans conformation (and would therefore represent the ground state), the authors skate over this difference, by suggesting that the “relaxed ground state of proteorhodopsin could be characterized by alternating conformations, resulting in the selective hydration of the intra- and extracellular cavities”. This explanation would appear to be a little unsatisfactory, since it would imply that there is a small energy barrier (relative to kT) for the conversion between forms and, if that were to be the case, then the protein samples prepared for cryo-EM would include a mixed population of the two states. They also identify differences in internal hydrogen bonding between the GPR and BPR structures and these bonds which would appear to be too strong to allow significant conversion between two states at physiological temperatures. This proposal requires further evidence before it can be confidently advanced and I would suggest to the authors that they present computational calculations (which might include simulation of a GPR D96N mutant) in order to confirm whether conversion between the two states is possible without retinal isomerisation. They should also make a detailed comment on the thermodynamics of the mechanism of conversion between the two states.

Authors: The Reviewer raises an interesting question regarding the seemingly low-energy barrier for the proposed conformational change. To address this issue, we performed multiple sets of MD simulations using lipid-embedded wild-type GPR. Indeed, we could show that the conversion between outward- and inward-open conformation is thermodynamically feasible at ambient temperature without retinal isomerization (Fig. S7). However, the change in conformation requires protonation or deprotonation of the primary proton donor E109. These results support our original observation regarding the two conformations in the ground state and provide a basic mechanism based on the protonation state of E109. We have elaborated our statements on this topic accordingly and included the new results in the revised version of the manuscript (see Fig. S7 and tracked changes)

Furthermore, since the conformational changes seem to be triggered by protonation or deprotonation of E109, it is unlikely to find a mixture of populations in the cryo-EM sample. Nevertheless, we assessed the flexibility of areas that would exhibit increased variability, as a result of two coexisting conformations, by analyzing the local resolution (Fig. S9) and employing 3D variability analysis (cryoSPARC). However, we found no evidence for increased variability, which suggests the presence of only one conformational state.

A great strength of this paper is that the authors present a structure for pentameric GPR. They claim that their structure “substantiates the functional relevance of GPR oligomerization”, however they do not explain this further. An oligomerisation interface (involving helices A, B and the N-terminal part of C) is identified, however these helices show only minimal changes in orientation between the ‘inward open’ and ‘outward open’ conformations. The functional relevance of the pentamer is therefore unclear, as it would be that the same range of movement of individual helices is permitted in both the monomeric and pentameric forms. There is no suggestion (or obvious mechanism) for cooperative behavior between protomers. The authors should make clear what they mean by the “functional relevance of GPR oligomerization”, explain what conformational changes are restricted/promoted in monomeric vs pentameric protein and identify potential molecular mechanisms which might make ion transport more efficient in the pentamer.

Authors: The authors thank the Reviewer for appreciating the strength of our GPR structure. The mentioned helices, i.e., A, B and N-terminal part of C, indeed stabilize the oligomerization interface. It is their strong involvement in these interactions that restricts their movement during conformational changes and indicates a function as rigid scaffold, against which the remaining helices can pivot. This resembles an established concept in the field of membrane transport proteins, which is commonly referred to as "rocking-bundle" mechanism (Bosshart & Fotiadis, *Subcell. Biochem.*, 2019). It is possible that this mechanism restricts the number of possible conformational changes and thus increases the probability of functionally relevant ones. We have adapted corresponding statements in the revised version of the manuscript to enhance the clarity of the presented concept (see tracked changes).

To provide further evidence for this mechanism, we have compared MD simulations of monomeric and pentameric GPR. We observed collectively higher root mean square fluctuations (RMSF) for residues in monomeric GPR than in the pentamer, which could indicate a regulatory function for oligomerization. The reduced flexibility in the pentamer likely originates from the stabilization provided by the oligomerization interface. Helices A-C, which are involved in this interface exhibit lower RMSF than helices D-G, which supports the scaffold function as proposed above. The scaffold function of helices A-C when engaging in the oligomerization interface and the increased rigidity of the GPR pentamer indicate a connection between the oligomeric state and the proton transport by GPR (see first paragraph in subchapter *Conformational Dynamics Allowing Selective Solvent Access*).

We have added the above mentioned new results (Fig. S6C) and their discussion (see tracked changes) in the revised version of the manuscript to corroborate our previous conclusions.

Minor suggestions to aid clarity:

Abstract

- "prototype" is the wrong word to use - "archetype" would be better, but consider rephrasing instead so that the word is not used.
- the phrase "inverse solvent accessibility" is unclear

Introduction

Page 3

- Line 3 - change "photochemical reaction centers, using a chlorophyll cofactor" to "chlorophyll-containing photochemical reaction centers"
- Line 12 - delete "respectively"
- Line 13 - change "and characterized as a light-driven proton pump" to "and was characterized as a light-driven proton pump"
- Line 14 - "prototype" is the wrong word to use here
- Line 16 - change "bionanotechnological applications" to "applications in bionanotechnology"
- Line 18 - change "focus of efforts to implement them as optogenetic tools" to "basis for the engineering of optogenetic tools"
- Line 21 - change "and a consequently increased proton translocation rate" to "and consequently, an increased proton translocation rate"

Page 4

- Line 2 - change "containing a covalently attached all-trans retinal, which forms the

characteristic Schiff base with a conserved lysine residue” to “containing a covalently bound all-trans retinal, which is conjugated to a conserved lysine residue via a Schiff base”

- Line 6 - change “crystal structures of the homolog BPR were solved” to “crystal structures of the homolog BPR have been solved”

- Line 8 - change “molecular insight in the GPR family has remained sparse” to “molecular insights into the GPR family have remained sparse”

- Line 11 - change “ambiguity about the oligomeric state prevented the preparation of a homogeneous protein sample” to “ambiguity about the oligomeric state has prevented the preparation of a homogeneous protein sample”

Results

Page 4

- Line 3 - delete “thus enabling structural characterization as similar to the native version as possible”

Page 5

- Line 3-14 - delete from “We compare the GPR structure...” to “functional importance of GPR oligomerization.” This paragraph is too long and too introductory.

Page 6

- Line 2 - replace “comprises” with “has”

- Line 12 - the abbreviation “OPM” should be explained

Page 7

- Fig 1 caption - authors should give the distances between the following residue pairs (either in the caption or as a table in the Supplementary Info): D98 and H76, D98 and D228, D228 and Y201, D228 and Y77.

Page 8

- Line 3 - replace “proteorhodopsin specific” with “proteorhodopsin-specific” (hyphen required)

Pages 10 and 11

The very long paragraph should be split into at least three smaller paragraphs

Page 12

- Line 19 - replace “which was also implicated” with “which is also implicated”

- Line 20 - replace “The

outline of the cavity is created by an accumulation of hydrophobic residues including” with “The

cavity is lined by hydrophobic residues including”

Page 13

- Line 2 - replace “This includes the formation of a hydrogen bond between E246 and S180, which

might contribute to stabilizing this closed conformation.” with “The hydrogen bond between E246 and S180 may stabilize this closed conformation.”

- Line 3 - new paragraph before “An inverse accessibility”

- Line 3 - replace “An inverse accessibility is observed in all BPR protomers, where the extracellular cavity” with “Conversely, in all BPR protomers, the extracellular cavity”
- Line 15 - new paragraph before ““An inverse accessibility”
- Line 16 - replace “favours” with “favors”
- Line 17 - delete “Interestingly”
- Line 18 - delete “also”
- Line 19 - replace “xanthorhodopsin (XR), interacting with the conserved” with xanthorhodopsin (XR), in which it interacts with the conserved”
- Line 20 - replace “Gloeobacter rhodopsin (GR), where the glutamate is substituted” with “Gloeobacter rhodopsin (GR), in which the glutamate is substituted”

Page 14

- Fig 3 caption - replace “E109 in the IC are indicated (orange).” with “E109 in the IC indicated in orange.”
- Fig 3 caption - delete “state with inverse solvent accessibility compared to GPR”
- Fig 3 caption - replace “E109 in the IC are indicated (pink).” with “E109 in the IC indicated in pink.”

Page 15

- Line 5 - delete “the solvent accessibility”
- Line 11 - replace “seem to serve as scaffold for the described motion of the other helices.” with “seem to serve as a scaffold against which the other helices may pivot.”
- Line 19 - replace “hypothesize” with “propose”

Authors: We thank the Reviewer for the proposed changes that have helped to improve the clarity of our manuscript. We have adapted them accordingly in the revised version (see tracked changes).

Reviewer #2:

This manuscript reports the first medium-high-resolution structure of Green Proteorhodopsin (GPR), the prototype of the large protein family of eubacterial proteorhodopsin proton pumps. The authors did employ a spectacular array of experimental and in silico methodology to unravel electron-microscopical images to submolecular resolution. Comparison with literature data of its distant cousin blue proteorhodopsin (BPR), apparently crystallized in a different functional state, results in a very workable model for the global functional mechanism of GPR and possibly ion-pumping rhodopsins in general. Still, a more detailed discussion of the differences in protomer structure between the pentameric state and other states, reported in the literature, would also be very informative. Further, the technical approach is well outlined, properly calibrated and persevered in depth, with careful data analysis and extensive in silico data reduction. The protein structure has been publicly deposited in the Protein Data Bank. Relevant literature has been properly covered. I would recommend that the authors take the following comments and suggestions into account.

Authors: The authors thank the Reviewer for appreciating the presented results. We have addressed the provided suggestions and have adapted the revised version of the manuscript accordingly.

1. The authors more or less claim that the pentameric state is the native one for GPR. Of course, this has not been analyzed for the native organism and in the membrane of the heterologous host *E. coli* several oligomeric states have been detected. A better wording would be that the pentameric state is the most prominent oligomeric state.

Authors: We thank the Reviewer for bringing up this important point. We have adapted corresponding statements in the revised version of the manuscript to reduce any ambiguity (see tracked changes).

2. Isolation and “clean” purification of GPR is essential in this research and merely referring to a recently published paper is not sufficient. A brief method section covering the essentials of these procedures should be included in the supplementary information. It is recommended to also include a SEC profile of the purified GPR.

Authors: The authors thank the Reviewer for this suggestion. We have included a detailed description of the experimental procedures and added an SDS-polyacrylamide gel and an analytical SEC profile of purified GPR in Supplementary Information (new Fig. S1). We refer to this new information in the Results and the Materials and Methods sections of the revised manuscript.

3. Eventually less than 20 % of the more than 3.6×10^6 particles was selected to proceed through the final processing and refinement software packages. It would be useful to put in whether the authors did find any evidence for other functional and/or oligomeric states in the rejected particles. Would it be an option to screen the rejects with a single optimized protomer structure in search for such other states ?

Authors: We thank the Reviewer for this suggestion. However, we have already exhausted the computational possibilities provided by the different processing software, without discovering significantly different conformational or oligomeric states. Specifically, we have performed multiple 2D and 3D classifications (Relion) as well as heterogeneous refinement runs (cryoSPARC) with various combinations of parameters and initial models to identify other possible oligomeric states or significantly different conformations. In addition, we have used 3D variability analysis (cryoSPARC) to identify any continuous conformational heterogeneity (i.e, potential flexibility of the protein density), which yielded no evidence for the presence of other functional states or conformations.

Due to the small size and the lack of prominent features, the protomer structure is not suited as a template for particle picking or as an initial model for 3D classification. Both procedures are based on algorithms that align particles by iterative incremental rotation and superposition. This requires the particles to have recognizable features, such as a prominent shape.

4. What is the advantage of using Cymal-5 over Cymal-4, and more general of using Cymal's instead of DDM ? Would the detergent selection influence the protomer interaction pattern or helix movements to a significant extent as compared to a membrane environment ?

Authors: In general, low-CMC detergents with large micelles add a large amount of disorder to protein particles. In single particle cryo-EM, this can prevent efficient alignment and thus significantly reduces the resolution of the final 3D reconstruction. To solve the GPR structure, we chose a relatively mild detergent, which does not disrupt the oligomeric state, with a minimal micelle size. Cymal detergents have lower aggregation numbers than typically used

n-alkyl maltosides, and thus form smaller micelles, increasing the probability of higher resolution reconstructions.

It is conceivable that different groups of detergents might favor particular protomer interaction patterns or allow different conformational changes. Importantly, we observed similar conformational changes to the ones we have proposed in the molecular dynamics (MD) simulation of lipid-embedded GPR. This shows that comparable helix movements can take place in both environments (i.e., detergent micelle and lipid bilayer).

5. According to ref. 20 a C-terminal his-tag still allows pentameric oligomerization. It would be valuable information for the rhodopsin field when any evidence can be extracted from the high-resolution structure as to whether a his-tag might still perturb the pentameric organization or helix movements.

Authors: Analysing the structure did not yield any strong evidence supporting the potential disruption of the GPR pentamer by addition of purification tags. Although a C-terminal His-tag is not directly attached to any helix involved in the formation of the oligomerization interface, i.e., helices A, B and the N-terminal part of helix C, potential interference with the oligomerization cannot be completely excluded. However, depending on the presence of a linker preceding the His-tag, the proximity of the C-terminus to the interface could facilitate limited interference. An N-terminal His-tag on the other hand seems more likely to cause perturbations as it is closer to the interface.

6. It would be useful to elaborate on the differences in protomer interaction between the pentameric state and the putative hexameric model in ref. 7 (Stone et al, 2013). Likewise, the possible structural consequences of monomerization would be very interesting, also in its potential bearing on the divergence with the NMR-based structure.

Authors: We thank the Reviewer for this interesting suggestion. Unfortunately, Stone et al. do not provide an atomic model of the hexamer, which would be required to analyze the specific interactions. In their study they used different spin-labeled residues in combination with EPR measurements to elucidate the orientation of the protomers within the oligomeric assembly and showed that helices A and B face towards the symmetry axis. This observation is confirmed by our GPR structure and demonstrates a similar assembly for the two oligomeric states.

We have assessed the consequences of monomerization by comparing MD simulations of pentameric and monomeric GPR. We observed collectively higher root mean square fluctuations (RMSF) for residues in monomeric GPR than in the pentamer. The reduced flexibility in the pentamer likely originates from the stabilization provided by the oligomerization interface. Helices A-C, which are involved in this interface exhibit lower RMSF than helices D-G, which supports the proposed scaffold function. We have elaborated more on this proposed function in the revised version of the manuscript and included the results of the simulations in Fig. S6C.

7. The entire procedure evolving into the final structural model involves a very complex process of software based selection and refinement. This of course also poses some risks. For instance:
a. In contrast to the BPR pentamer, all subunits of the GPR pentamer come out identical.

Can the authors exclude that this is an unfortunate consequence of the very complex in silico processes.

b. The resolution is, as expected, not evenly distributed over the entire protein, with about 2.9 Å in the best resolved regions. Does this include the chromophore and binding pocket residues? For instance, the electron density profile around the chromophore (Fig. S1) could suggest a mixture of an all-trans and a 13-cis configuration!? Further, I presume that water molecules and H-bonds have not been resolved? Nonetheless, such ambiguities will most likely not downgrade the global functional mechanism proposed in movie S1.

Authors: We thank the Reviewer for bringing up these questions and can confirm that we have been aware of these possible pitfalls. Throughout the processing of our data we have thus implemented corresponding control measures as outlined below.

A refinement process without imposed C_5 symmetry was performed, which yielded a practically identical structure, albeit with lower resolution (~3.6 Å). This indicates no significant introduction of symmetry-related bias.

We thank the reviewer for pointing out that the local resolution of the chromophore binding pocket was not clearly illustrated. Accordingly, we have updated Figure S7 (now Fig. S9) in the revised version of the manuscript to include a cross-section displaying the local resolution of the binding pocket, with the retinal highlighted. The new figure nicely shows that the local resolution of the chromophore and the surrounding binding pocket is around 2.6-3.0 Å. While the presence of a small percentage of 13-cis retinal cannot be completely excluded, it is significantly more likely to contain mainly all-trans retinal, based on a better fit into the density. Finally, as the Reviewer correctly assumes, the resolution of the presented structure did not allow modeling of water molecules or H-bonds.

Reviewer #3:

The article by Hirschi et al., describes the cryo-EM structure of a proteorhodopsin and decipher novel insights into the functioning of this light driven proton pump. Using the structural template, they discuss how conformational changes associated with various transmembrane helices facilitate the pentameric assembly of this protein. While the overall structural features are interesting and provide important insights, the lack of functional data to corroborate the structural findings somewhat limit the enthusiasm for this manuscript. For example, Figure 3 and related discussion deals with how various residues and their interactions may govern proton release and translocation cycle, but experimental data in the form of site-directed mutagenesis to corroborate these observations is not presented. Even if these data are not straightforward to generate, at least a more extensive effort should be taken to discuss this in the context of existing literature on other members of this family. The method section is way too brief and does not sufficiently describe the details of protein isolation and characterization. Overall, this is an interesting manuscript that should be published after some critical revision to better connect the structural observations with mechanistic aspects.

Authors: We thank the Reviewer for appreciating the presented work. To support our structural findings, we have produced and analyzed the effect of multiple uncharacterized GPR variants to corroborate our conclusions. Eight mutants (K58A, W59A, Y96F, E143A, S180A, Y209F, Y224F, E246A) were prepared by site-directed mutagenesis and their proton

pumping function was analysed after expression in *E. coli* cells using a photoactivity assay. These residues are located at the entrance and exit of the proposed proton translocation pathway, and were observed to form potentially important interactions in our structure. We could show that the majority of mutations diminished the proton pumping activity, demonstrating their importance for GPR function. Remarkably, we have also found mutations (S180A and E246A) that increase the efficiency of the proton transport. We concluded that the interaction between those two residues may have evolved to regulate the photocycle dynamics of GPR. We have discussed the results in an additional subchapter of the Results section (*Functional Relevance of Residues at the Entrance and Exit of the Proton Translocation Pathway*) and included a new figure (Fig. 4) in the revised version of the manuscript.

Furthermore, we have performed molecular dynamics simulations, which support our mechanistic conclusions regarding the proposed conformational changes and scaffold function of helices A-C in pentameric GPR. Specifically, the simulations have revealed that the change between outward-open and inward-open states depends on the protonation state of the primary proton donor E109. We have discussed these new results in the revised version of the manuscript and included new figures (Fig. S6 and S7).

Finally, we have included a more detailed description of the experimental procedures and added an SDS-polyacrylamide gel and an analytical SEC profile of purified GPR in the Supplementary Information (new Fig. S1). We refer to this new information in the Results and in the Materials and Methods sections of the revised manuscript.

REVIEWER COMMENTS

Reviewer #1 (Remarks to the Author):

In this paper, Hirschi and coworkers report a cryo-EM structure of the green-light absorbing proteorhodopsin (GPR). The resolution achieved by this method is undoubtedly significant, especially for a protein of this size in its apparent native oligomeric state. A great strength of this paper is that the authors present a structure for pentameric GPR.

In response to the comments from me and the other reviewers, the authors have analysed their structural data in more detail and have performed additional experiments. In particular, using MD simulations they have been able to comment on the energetic barriers between the inward-open and outward-open conformations of the GPR ground state. I am satisfied that in its revised form, the article meets the high standards expected for Nature Communications.

Some minor suggestions to aid clarity:

- Page 5, line 156 - change "even in absence of an imposed five-fold symmetry constraint (C1) during refinement." to "even in the absence of a five-fold symmetry constraint (C1) imposed during refinement."
- Page 7, line 237 - change "proteorhodopsin specific histidine" to "proteorhodopsin-specific histidine"
- Page 11, line 366 - change "with the exception of a glutamine found at position 54 in BPR instead of an arginine" to "with the exception of a glutamine instead of an arginine at position 54 in BPR"
- Page 13, line 449 - replace "interacting with the conserved K58" with "that interacts with the conserved K58"
- Page 15, line 490 - replace "The observed interactions between residues" with "The interactions observed between residues"
- Page 17, line 559 - delete "by the described helix disturbances"
- Page 17, line 565 - start a new paragraph before "In order to test"
- Page 18, line 590 - start a new paragraph before "Furthermore, we have specifically"

Reviewer #2 (Remarks to the Author):

The authors have done an excellent job in revising the manuscript according to the suggestions of the reviewers. The response to the reviewer comments and the new information inserted definitely further improve the paper and support the global switch-mechanism proposed for the pump activity. I still have a small number of final questions and comments:

1. The mutant data included is very informative. It should be noted, however, that the Y96F mutant has been reported before (Y95F in Kim et al (2008) BBA 1777:504). The authors should include a reference to this paper. Furthermore I recommend to include any spectral effects of the mutations in Table S1.
2. It is not clear to me in the new methods section on the photo-activity analysis how the pump activity actually was measured. Please state whether this was taken as the initial rate of the change in the proton concentration upon illumination or as the maximal difference between the proton concentration before and during or after illumination. The latter of course not only depends on the pump activity, hence is less reliable.
3. According to Fig. S6 the dynamics in the pentamer are a bit more restrained than in the monomer, but the differences are not dramatic. Can this really explain the much slower late-stage photokinetics of the oligomer (Hussain et al, ref. 36) or the significant differences between the pentamer subunit cryostructure and the monomer NMR structure (Reckel et al, ref. 18) ? The literature data were obtained in DDM and/or DHPC, both quite mild detergents. If that nevertheless would be an important factor, an effect of Cymal-4 cannot be excluded. The authors did not really discuss these topics, but they might have relevant thoughts about these issues, worthwhile to share with the reader in the discussion sections.

4. The authors propose a switch mechanism mainly regulated by protonation of E109. I find this a bit difficult to follow. Is the following sequel what the authors have in mind?: In the ground state E109 is protonated and the extracellular "channel" is open, but E143 is protonated. Upon photoisomerization the Schiff base proton is taken up by the D98 counterion and channeled into the extracellular cavity, eventually releasing and replacing the proton on E143. Upon entering the reprotonation and thermal re-isomerization stage, the E109 proton is released and eventually taken up by the Schiff base, and the deprotonated E109 opens up the intracellular channel and locks the extracellular channel, until it is reprotonated and the ground state is reached again? If this is correct, it may be sensible to insert such a textual segment somewhere.

Reviewer #3 (Remarks to the Author):

The authors have satisfactorily addressed the points raised on the original manuscript, and therefore, I recommend publication of this manuscript in Nature Communications.
Arun K. Shukla, Ph.D.

Reviewer #4 (Remarks to the Author):

The study by Hirschi et al is obviously important to the field of microbial rhodopsins -- green proteorhodopsin (GPR) is superior to blue proteorhodopsin (BPR) in almost every aspect with respect to function, yet has been notoriously difficult to obtain an X-ray crystal structure. However, the broad impact of the solution of this structure is somewhat overstated, and it's largely not the fault of the authors -- the field of structural characterization of microbial rhodopsins has simply made huge strides since the NMR solution of GPR, in particular the recent series of solutions of X-ray crystal structures of KR2 and Gloeobacter rhodopsin, both of which have more promise in optogenetics applications. Thus, the lack of comparison and proper contextualization of the current study with other microbial rhodopsins (and by extension, the exclusive comparison to BPR) weaken the overall impact of the paper. By addressing these issues, the manuscript should be a fine addition to Nature Communications. Specific suggestions below.

1) With respect to comparison to other microbial rhodopsins:

- 1a) bacteriorhodopsin (bR) is barely mentioned. The authors should note that they are not the first group to show that deprotonation of the proton donor opens up the intercellular portion of a microbial rhodopsin. Duan, Facciotti, and coworkers (Structure 2013) elegantly demonstrated that deprotonation of the proton donor, D96, acted as a latch mechanism in which the proton uptake segment of bR underwent large-scale movement of TM helices within 50 ns in every single (of dozens) molecular dynamics (MD) simulation, regardless of the photointermediate structure that was used.
- 1b) In addition, the proton release group (PRG) of bR, to date, is the only microbial rhodopsin that has been definitively identified (Gerwert, Bashford, et al., JMB 2001 and Onufriev, Bashford, et al., JMB 2003) -- there has been no experimental confirmation that E142 is the PRG in GPR. While the results presented here are compelling towards identification of E142 as the PRG, it requires additional experiments (e.g., FTIR) to validate this hypothesis.
- 1c) KR2, Gloeobacter rhodopsin, and mastR (to name a few examples) have all had X-ray crystal structures solved in the past 3 years, all in a pentameric oligomer. A comparison to these other microbial rhodopsins would be extremely beneficial.

2) With respect to previous functional studies on GPR, more discussion should be devoted towards the extension of helix E (E*) in the cryo-EM structure. In particular, how does this extension potentially affect the behavior of the EF loop, which has been shown to be a critical player in the GPR photocycle (Hussan, Han et al, Angew. Chemie Intl. Ed. 2013)? In addition, the A178R mutation is another color-tuning variant of GPR that could be affected by the E* extension. The timescale and different

protonation states employed in the MD simulations could potentially provide some insights into this matter.

3) Reviewer #1 made an excellent point that has still not been fully addressed: there needs to be a well thought-out justification for why such a low thermodynamic barrier exists between the inward- and outward-facing states of GPR. The observed behavior from the MD simulations is simply an artifact (see comments above about the Duan study): there is no physically relevant protonation state of GPR in the ground state (all-trans retinal) with a deprotonated E108. Thus, conclusions about the functional relevance of this observed phenomena in the MD simulations come across as very speculative.

4) In addition, the connection between the pentameric state of GPR and function has not been definitively shown in this study; it is premature to claim this is so. The authors should tone down their claims and pull extensively from studies that have more definitively shown this connection (Hussain, Han, et al., Ref. 36, Idso, Han, et al., J. Phys. Chem. B. 2019, and Han, Han, et al. Biophys. J. 2020).

5) Several issues need to be addressed and additional analysis needs to be carried out on the MD simulations:

5a) Hydrogen mass repartitioning (timestep of 4 fs) in a membrane system is still very experimental, as it could lead to system instabilities. Please provide deuterium order parameter calculations of the lipids to show that the system is stable.

5b) Internal hydration of the intracellular half of PR via deprotonation of the proton donor has already been shown via MD simulations for BPR -- Jun and Mertz (Ref. 54) and Faramarzi et al., Biophys. J. 2018). In addition, these studies also showed that the extracellular half of BPR becomes hydrated after isomerization of retinal -- it would be beneficial to compare to that particular analysis as well.

5c) The MD simulations mainly focused on RMSF and distances between key residue pairs. What does the internal hydration of GPR look like as a function of protonation state?

5d) How do rotamers of W34 and H75 change for the monomer versus the pentamer form of GPR?

5e) Why was there no analysis of S180 and E246, considering that their respective point mutations led to an increase in photoactivation?

5f) Psfgen in VMD cannot generate a force field. Where was the retinal force field derived from? This is important, as Zhu, Feller, et al. JACS 2013 showed that subtle changes to the retinal force field can have significant effects on the behavior of different segments of the chromophore (notably the beta-ionone ring and the polyene chain), which in turn can affect larger-scale conformational dynamics of rhodopsin.

5g) Please specify the cutoff used for non-bonded forces.

5h) Why was the FG loop not modeled in the simulations? There are several codes (e.g., MODELLER) that could be used to model in the FG loop. This segment may not be important to oligomerization, since it is distal from the protomer-protomer interface, but could affect some of the inward- to outward-facing conformational changes associated with the rearrangement of helix F and G that were proposed in Figure 5.

Reviewer #1:

In this paper, Hirschi and coworkers report a cryo-EM structure of the green-light absorbing proteorhodopsin (GPR). The resolution achieved by this method is undoubtedly significant, especially for a protein of this size in its apparent native oligomeric state. A great strength of this paper is that the authors present a structure for pentameric GPR.

In response to the comments from me and the other reviewers, the authors have analysed their structural data in more detail and have performed additional experiments. In particular, using MD simulations they have been able to comment on the energetic barriers between the inward-open and outward-open conformations of the GPR ground state. I am satisfied that in its revised form, the article meets the high standards expected for Nature Communications.

Authors: The authors thank the Reviewer for appreciating our efforts and work.

Some minor suggestions to aid clarity:

- Page 5, line 156 - change "even in absence of an imposed five-fold symmetry constraint (C1) during refinement." to "even in the absence of a five-fold symmetry constraint (C1) imposed during refinement."
- Page 7, line 237 - change "proteorhodopsin specific histidine" to "proteorhodopsin-specific histidine"
- Page 11, line 366 - change "with the exception of a glutamine found at position 54 in BPR instead of an arginine" to "with the exception of a glutamine instead of an arginine at position 54 in BPR"
- Page 13, line 449 - replace "interacting with the conserved K58" with "that interacts with the conserved K58"
- Page 15, line 490 - replace "The observed interactions between residues" with "The interactions observed between residues"
- Page 17, line 559 - delete "by the described helix disturbances"
- Page 17, line 565 - start a new paragraph before "In order to test"
- Page 18, line 590 - start a new paragraph before "Furthermore, we have specifically"

Authors: We thank the Reviewer for the suggestions.

Reviewer #2

The authors have done an excellent job in revising the manuscript according to the suggestions of the reviewers. The response to the reviewer comments and the new information inserted definitely further improve the paper and support the global switch-mechanism proposed for the pump activity. I still have a small number of final questions and comments:

Authors: We thank the Reviewer for appreciating our revision and added results.

1. The mutant data included is very informative. It should be noted, however, that the Y96F mutant has been reported before (Y95F in Kim et al (2008) BBA 1777:504). The authors should

include a reference to this paper. Furthermore I recommend to include any spectral effects of the mutations in Table S1.

Authors: We thank the Reviewer for bringing this finding to our attention. We have included the chromophore absorption maxima in Table S1 and cited the missing reference.

2. It is not clear to me in the new methods section on the photo-activity analysis how the pump activity actually was measured. Please state whether this was taken as the initial rate of the change in the proton concentration upon illumination or as the maximal difference between the proton concentration before and during or after illumination. The latter of course not only depends on the pump activity, hence is less reliable.

Authors: We have adapted this section to specify that we measured the maximal difference of the proton concentration before and at the end of the illumination. When carefully controlling external factors such as starting pH, bacterial cell density etc., this procedure should provide data to reliably compare the activity of the GPR mutants.

3. According to Fig. S6 the dynamics in the pentamer are a bit more restrained than in the monomer, but the differences are not dramatic. Can this really explain the much slower late-stage photokinetics of the oligomer (Hussain et al, ref. 36) or the significant differences between the pentamer subunit cryostructure and the monomer NMR structure (Reckel et al, ref. 18) ? The literature data were obtained in DDM and/or DHPC, both quite mild detergents. If that nevertheless would be an important factor, an effect of Cymal-4 cannot be excluded. The authors did not really discuss these topics, but they might have relevant thoughts about these issues, worthwhile to share with the reader in the discussion sections.

Authors: We agree with the Reviewer that the RMSF differences are not dramatic. However, they are systematically observed in each replica of the MD simulations. We propose that the slower late-stage photocycle kinetics probably result from a combined effect and not solely from the increase in rigidity. As for the significant differences between the monomer structures of GPR, they are most likely a result of the structure solution process, i.e. imposing multiple structural restraints on the NMR models, and the overall lower quality of the NMR data. The significantly higher RMSD between the cryo-EM structure and the NMR structures compared to the BPR crystal structure, despite the lower amino acid sequence identity with the latter, indicates some inconsistency within the NMR structure.

Hussain et al. demonstrated the functional effects of GPR oligomerization. A slower M-state decay was observed for the oligomer, independently of the detergent (DDM or DPC), which indicates that protein-protein interactions are more relevant than those between protein and surfactant. Based on their SEC data, it is evident that DPC is distinctly harsher than DDM, disrupting the oligomeric state and resulting in a significant increase of the monomer population. Purifications of GPR using Cymal-4 result in a chromatographic profile that is more similar to that of DDM, indicating that DPC is also harsher than Cymal-4. It is therefore reasonable to expect no significant effects between the detergents used here and in the literature. In a follow-up study (Idso et al., J. Phys. Chem. B. 2019), a more diverse range of detergents was tested with a similar outcome, supporting the conclusion that the effects of the oligomeric state are dominant compared to the detergent environment.

We have expanded the discussion on the functional effects of GPR oligomerization with a more detailed comparison to the mentioned references (e.g., end of page 18).

4. The authors propose a switch mechanism mainly regulated by protonation of E109. I find this a bit difficult to follow. Is the following sequel what the authors have in mind?: In the ground state E109 is protonated and the extracellular “channel” is open, but E143 is protonated. Upon photoisomerization the Schiff base proton is taken up by the D98 counterion and channeled into the extracellular cavity, eventually releasing and replacing the proton on E143. Upon entering the reprotonation and thermal re-isomerization stage, the E109 proton is released and eventually taken up by the Schiff base, and the deprotonated E109 opens up the intracellular channel and locks the extracellular channel, until it is reprotonated and the ground state is reached again? If this is correct, it may be sensible to insert such a textual segment somewhere.

Authors: We thank the Reviewer for pointing out this ambiguity. In fact, our results describe two main findings: i) the structural variations between the GPR outward-open and the BPR inward-open conformations, which allow selective solvent access to the intra- and extracellular cavities and ii) the conformational dynamics and hydration of the intracellular half channel during MD simulations, which are triggered by a change in the protonation state of E109. In the revised version of the manuscript, we have divided the corresponding chapter in two parts to better present these results.

We agree that the physiological relevance of E109 as a switch has not been discussed appropriately. Thus, we have now carefully revisited this issue in the context of the GPR photocycle. Opening and hydration of the intracellular half channel (see new Figs. 6 and S10) would be triggered after the primary proton donor E109 has donated a proton to the Schiff base during the M-N transition. The subsequent influx of water enables reprotonation of E109 and results in closing of the cavity, which is proposed to occur between N- and O-intermediates, just before returning to the ground state.

We have elaborated our proposed mechanism in the revised manuscript.

Reviewer #3

The authors have satisfactorily addressed the points raised on the original manuscript, and therefore, I recommend publication of this manuscript in Nature Communications.

Arun K. Shukla, Ph.D.

Authors: The authors thank the Reviewer for recommending our manuscript for publication.

Reviewer #4

The study by Hirschi et al is obviously important to the field of microbial rhodopsins -- green proteorhodopsin (GPR) is superior to blue proteorhodopsin (BPR) in almost every aspect with respect to function, yet has been notoriously difficult to obtain an X-ray crystal structure. However, the broad impact of the solution of this structure is somewhat overstated, and it's largely not the fault of the authors -- the field of structural characterization of microbial rhodopsins has simply made huge strides since the NMR solution of GPR, in particular the recent series of solutions of X-ray crystal structures of KR2 and *Gloeobacter* rhodopsin, both of which have more promise in optogenetics applications. Thus, the lack of comparison and proper contextualization of the current study with other microbial rhodopsins (and by extension, the exclusive comparison to BPR) weaken the overall impact of the paper. By addressing these

issues, the manuscript should be a fine addition to Nature Communications. Specific suggestions below.

Authors: We thank the Reviewer for appreciating the presented manuscript and for the constructive suggestions below to improve its overall impact.

1) With respect to comparison to other microbial rhodopsins:

1a) bacteriorhodopsin (bR) is barely mentioned. The authors should note that they are not the first group to show that deprotonation of the proton donor opens up the intercellular portion of a microbial rhodopsin. Duan, Facciotti, and coworkers (Structure 2013) elegantly demonstrated that deprotonation of the proton donor, D96, acted as a latch mechanism in which the proton uptake segment of bR underwent large-scale movement of TM helices within 50 ns in every single (of dozens) molecular dynamics (MD) simulation, regardless of the photointermediate structure that was used.

Authors: The authors are very grateful to the Reviewer for indicating this missing reference to the mentioned BR study. Wang et al. assessed the structural effects of deprotonating the primary proton donor D96 (analogous to E109 in GPR) by MD simulations. They concluded that deprotonation of D96 triggers the opening and the hydration of the cytoplasmic half channel of BR, which seems to be independent of retinal isomerization. For comparison, we have analyzed the hydration patterns of the intra- and extracellular half channels in our GPR simulations with different protonation states for E109 and E143 (new Figs. 6 and S10). We could observe hydration of the intracellular half channel when E109 is deprotonated, similar to the findings in Wang et al. In addition, we discovered a corresponding effect for E143. Interestingly, deprotonation of E143 and hydration of the extracellular half channel is not accompanied by comparably large conformational changes (Fig. S9).

In conclusion, our results agree well with the study of Wang and colleagues and provide additional analysis of the protonation states of E143. We have added a discussion of this analysis and the comparison with their study in the revised version of the manuscript.

1b) In addition, the proton release group (PRG) of bR, to date, is the only microbial rhodopsin that has been definitively identified (Gerwert, Bashford, et al., JMB 2001 and Onufriev, Bashford, et al., JMB 2003) -- there has been no experimental confirmation that E142 is the PRG in GPR. While the results presented here are compelling towards identification of E142 as the PRG, it requires additional experiments (e.g., FTIR) to validate this hypothesis.

Authors: We thank the Reviewer for pointing this out. We have changed statements regarding E143 to indicate that it is only the proposed proton release group. In light of the newly observed hydration patterns of the extracellular half channel depending on the protonation state of E143 (new Figs. 6 and S10), we have added that our results could support the assignment of E143 as the proton release group in GPR.

1c) KR2, Gloeobacter rhodopsin, and mastR (to name a few examples) have all had X-ray crystal structures solved in the past 3 years, all in a pentameric oligomer. A comparison to these other microbial rhodopsins would be extremely beneficial.

Authors: The authors agree that comparison of the GPR structure to other microbial rhodopsins is valuable. We have initially refrained from detailed structural comparison with BR and XR since Ran et al., Acta Cryst. D, 2013 have already extensively compared these structures to

their BPR crystal structure. However, the structures of new microbial light-driven proton pumps have been solved since then. In particular, those of *Exiguobacterium sibiricum* rhodopsin (ESR) and *Gloeobacter* rhodopsin (GR). We have now included a comparison between these structures and GPR, focusing on the main functional residues involved in proton translocation (new Fig. S7A). Unfortunately, the oligomeric structures of neither ESR (whose native oligomeric state is currently unknown) nor GR (which has been demonstrated to form pentamers) have been solved, precluding the comparison of the oligomerization interface. Although several microbial rhodopsins have been demonstrated to assemble into pentamers (Shibata, Sci. Rep., 2018), KR2 is the only one beside BPR whose structure has been solved in the pentameric state. Due to the difference in function (KR2 is a sodium pump), we have focused on the analysis and comparison of the oligomerization interface between GPR and KR2 (new Fig. S7B). We have not included a comparison with the chloride pump mastR, because its structure was solved as a trimer (Besaw, JBC, 2020).

2) With respect to previous functional studies on GPR, more discussion should be devoted towards the extension of helix E (E*) in the cryo-EM structure. In particular, how does this extension potentially affect the behavior of the EF loop, which has been shown to be a critical player in the GPR photocycle (Hussain, Han et al, Angew. Chemie Intl. Ed. 2013)? In addition, the A178R mutation is another color-tuning variant of GPR that could be affected by the E* extension. The timescale and different protonation states employed in the MD simulations could potentially provide some insights into this matter.

Authors: Hussain et al. have demonstrated using ODNP and EPR that part of the E-F loop forms an α -helical segment and could resolve the relative orientation of its residues. Both of these findings are confirmed by our cryo-EM structure. Furthermore, they have looked at the dynamics of this α -helical segment and its surrounding water upon photoactivation. The change in side chain and hydration dynamics observed for particular loop residues upon photoactivation are not comparable to our results as we do not employ photoisomerization of the retinal chromophore during our simulations.

As for how the α -helical structure affects the behavior of the E-F loop, it is likely that the secondary structure restrains the movement of helices E and F and thus regulates their dynamics during the photocycle. Disturbing this structure leads to effects as described for the color-tuning mutation A178R. The latter has been shown to alter the absorption maximum of GPR and the pKa of the primary proton donor, which is surprising given the residue's remote position relative to the chromophore. In the cryo-EM structure the side chain of A178 faces the center of the protein. Substitution with a bulky residue, such as an arginine, would thus significantly distort the local structure due to the generation of steric clashes. To accommodate the large residue, the E-F loop helix would need to partially unfold. This supports a proposed model for the structural consequence of the A178R mutation (Mehler et al., Biophys. J., 2013), which provides an interaction pathway from the distorted E-F loop to the retinal binding pocket and which explains the significant change in the pKa of the primary proton donor and the chromophore absorption maximum.

We have added a discussion of the E-F loop structure and the effects of the A178R mutation in context of the relevant literature (page 6, please see tracked changes).

3) Reviewer #1 made an excellent point that has still not been fully addressed: there needs to be a well thought-out justification for why such a low thermodynamic barrier exists between the

inward- and outward-facing states of GPR. The observed behavior from the MD simulations is simply an artifact (see comments above about the Duan study): there is no physically relevant protonation state of GPR in the ground state (all-trans retinal) with a deprotonated E108. Thus, conclusions about the functional relevance of this observed phenomena in the MD simulations come across as very speculative.

Authors: The conformational change of the intracellular half channel triggered by altering the protonation state of E109 does not seem to be an artifact since i) several studies have demonstrated similar results (Wang et al., *Structure*, 2013 and Faramarzi et al., *Biophys. J.*, 2018), and ii) employing the protonation state of E109 as a signal to induce cytoplasmic proton uptake by hydration of the channel would be a reasonable physiological mechanism. We agree that the biological relevance of this switch has not been discussed appropriately in the previous version of our manuscript. Thus, we have now carefully revisited this issue in the context of the GPR photocycle. We propose that the observed changes in conformation and hydration of the half channels describe transitions between late-stage photocycle intermediates concurrent with the cytoplasmic uptake of a proton. Specifically, the opening of the intracellular half channel would be triggered after E109 has donated its proton to the Schiff base during the M-N transition, which allows hydration of the channel and reprotonation of E109. Reprotonation of E109, which is proposed to occur between N- and O-intermediates, induces the closing of the intracellular half channel, reestablishing the ground state conformation.

We have updated the discussion of this proposed mechanism in the revised version of the manuscript.

4) In addition, the connection between the pentameric state of GPR and function has not been definitively shown in this study; it is premature to claim this is so. The authors should tone down their claims and pull extensively from studies that have more definitively shown this connection (Hussain, Han, et al., Ref. 36, Idso, Han, et al., *J. Phys. Chem. B.* 2019, and Han, Han, et al. *Biophys. J.* 2020).

Authors: The studies by Hussain et al. and Idso et al. both demonstrate the effects of GPR oligomerization on different aspects of the protein function. In particular, a decrease in the pKa value of the primary proton acceptor D98 and a slower decay of the M intermediate were observed in the oligomer compared to the monomer. We have added a discussion on how the lack of the W35-H76 interaction helps to explain the change in pKa and how the increased rigidity of the pentamer might contribute to slowing down the photocycle kinetics. The missing references have been added and we have adapted our statements regarding the observed functional effect of GPR oligomerization.

5) Several issues need to be addressed and additional analysis needs to be carried out on the MD simulations:

5a) Hydrogen mass repartitioning (timestep of 4 fs) in a membrane system is still very experimental, as it could lead to system instabilities. Please provide deuterium order parameter calculations of the lipids to show that the system is stable.

Authors: Balusek et al., *J. Chem. Theory Comput.*, 2019 have shown that the hydrogen mass repartitioning approach can be used for membrane simulations. We have added the reference to the Methods section. To validate our approach, we calculated the deuterium order parameter of the lipids used in the simulations (see Figure below) and could confirm the stability of the system.

5b) Internal hydration of the intracellular half of PR via deprotonation of the proton donor has already been shown via MD simulations for BPR – Jun and Mertz (Ref. 54) and Faramarzi et al., *Biophys. J.* 2018). In addition, these studies also showed that the extracellular half of BPR becomes hydrated after isomerization of retinal -- it would be beneficial to compare to that particular analysis as well.

5c) The MD simulations mainly focused on RMSF and distances between key residue pairs. What does the internal hydration of GPR look like as a function of protonation state?

Authors: The authors thank the Reviewer for bringing these reports to our attention. We have added the analysis of the hydration for the intra- and extracellular half channels of GPR for all simulations with the different E109 and E143 protonation states (new Figs. 6 and S10). We could observe a similar hydration pattern as Faramarzi et al. and Feng et al. when the protonation state of E109 is changed. In addition, we have also observed a significant increase in hydration of the extracellular half channel upon deprotonation of E143 (new Figs. 6 and S10). We have added a discussion of these findings and comparison to the relevant literature in the revised version of the manuscript.

5d) How do rotamers of W34 and H75 change for the monomer versus the pentamer form of GPR?

Authors: We observed an increase in the number of rotamers sampled by W35 and H76 for the GPR pentamer compared to the monomer. In particular the W35 rotamer, which allows the cross-protomer interaction with H76 is only found in the pentamer. We have discussed these findings and added a new figure (new Fig. S6) illustrating the different conformations sampled during the MD simulation of the GPR pentamer and monomer.

5e) Why was there no analysis of S180 and E246, considering that their respective point mutations led to an increase in photoactivation?

Authors: We analyzed the interactions of S180 and E246 during the MD simulations and found that hydrogen bonding between the two residues occurred more frequently in simulations where E109 was protonated. The average hydrogen bonding frequencies over the entire simulation period were about 83% when E109 was protonated and 50% when it was deprotonated. This observation supports our hypothesis that this interaction preferentially stabilizes the conformation with a closed intracellular half channel, which might slow down the overall

photocycle. We have added the results of this analysis and discussed it in the revised version of the manuscript (e.g., on page 21).

5f) Psfgen in VMD cannot generate a force field. Where was the retinal force field derived from? This is important, as Zhu, Feller, et al. JACS 2013 showed that subtle changes to the retinal force field can have significant effects on the behavior of different segments of the chromophore (notably the beta-ionone ring and the polyene chain), which in turn can affect larger-scale conformational dynamics of rhodopsin.

Authors: The Reviewer brings up an important point, that the choice of retinal force field is crucial for the outcome of the simulations. For our simulations, the parameters were taken from a previous MD study of BPR. To improve the clarity, we have rephrased the corresponding sentence in the Methods section.

5g) Please specify the cutoff used for non-bonded forces.

Authors: We used the standard cut-off of 12 Å, with a switching distance at 10 Å. This has been clarified in the Method section.

5h) Why was the FG loop not modeled in the simulations? There are several codes (e.g., MODELLER) that could be used to model in the FG loop. This segment may not be important to oligomerization, since it is distal from the protomer-protomer interface, but could affect some of the inward- to outward-facing conformational changes associated with the rearrangement of helix F and G that were proposed in Figure 5.

Authors: We decided not to model the FG loop for the simulations since the loop is very flexible (due to three glycines) and the accurate modeling of loops remains a challenging problem. Thus, we only simulated the experimental structure, in order to avoid adding potential errors from the loop modeling.

REVIEWERS' COMMENTS

Reviewer #2 (Remarks to the Author):

The authors have appropriately reacted to my remaining items 1 and 4, but I still have my reserve about their response to items 2 and 3.

Re item 2: The maximal difference in the proton concentration upon illumination of a proton pump expressing cell population of course also depends on cellular factors like proton leakage, intracellular pH buffering, general membrane properties etc. This cannot be fully controlled between cell populations expressing different mutants. In addition, the molar absorbance of the mutants may vary. Hence, the error in the proton pump data presented in the bar chart in figure 4C is certainly underestimated. Nevertheless, this will not strongly affect the overall pattern, and therefore will not affect the for this paper relevant conclusions.

Re item 3: Hussain et al. also demonstrated that the M-decay rate of the GPR oligomer in the more aggressive detergent DPC is significantly higher than in DDM, and the GPR monomer actually is very unstable in DPC. Hence, the detergent environment clearly does have an influence. The molecular dynamics simulation, which does not reveal dramatic differences in dynamics of the monomer and the pentamer, is performed on a lipid-embedded model, however. Therefore, one interpretation might be that in a membrane-like environment there is less difference in dynamical space between the pentamer and the oligomer than in a detergent environment. Rather than implying that the pentamer is the functional state, it raises the question if and how the native cell can exploit the differences in functionality between the pentamer and the monomer.

Reviewer #4 (Remarks to the Author):

The authors have addressed all of my concerns -- the manuscript is much improved.

Reviewer #2:

The authors have appropriately reacted to my remaining items 1 and 4, but I still have my reserve about their response to items 2 and 3.

Re item 2: The maximal difference in the proton concentration upon illumination of a proton pump expressing cell population of course also depends on cellular factors like proton leakage, intracellular pH buffering, general membrane properties etc. This cannot be fully controlled between cell populations expressing different mutants. In addition, the molar absorbance of the mutants may vary. Hence, the error in the proton pump data presented in the bar chart in figure 4C is certainly underestimated. Nevertheless, this will not strongly affect the overall pattern, and therefore will not affect the for this paper relevant conclusions.

Authors: We thank the Reviewer for the input and for elaborating the potential variables of the measurements. We have reanalysed the data to compare the maximal $[H^+]$ differences and initial rates for the green-light absorbing proteorhodopsin (GPR) mutants. As predicted by the Reviewer the overall pattern is very similar and consequently does not change our final conclusion.

Re item 3: Hussain et al. also demonstrated that the M-decay rate of the GPR oligomer in the more aggressive detergent DPC is significantly higher than in DDM, and the GPR monomer actually is very unstable in DPC. Hence, the detergent environment clearly does have an influence. The molecular dynamics simulation, which does not reveal dramatic differences in dynamics of the monomer and the pentamer, is performed on a lipid-embedded model, however. Therefore, one interpretation might be that in a membrane-like environment there is less difference in dynamical space between the pentamer and the oligomer than in a detergent environment. Rather than implying that the pentamer is the functional state, it raises the question if and how the native cell can exploit the differences in functionality between the pentamer and the monomer.

Authors: We agree that the detergent environment is certainly important for the function of GPR. However, as soon as the detergent is too harsh it is very difficult to discern between the effects of disrupting the oligomeric state and the effects of protein-detergent interactions on the photocycle. In this study we have only investigated the dynamics of GPR in the ground state and when changing protonation states of important functional residues. To fully understand the functional impact of GPR oligomerization on the photocycle, extensive molecular dynamics simulations that model the photoisomerization process would need to be performed for the pentamer and monomer. This is indeed an interesting aspect for future studies, but lies outside the scope of the current manuscript.

Reviewer #4:

The authors have addressed all of my concerns -- the manuscript is much improved.

Authors: We thank the Reviewer for the valuable suggestions that helped strengthen our manuscript.